# Redundancy of macrobenthic functional traits boosts resilience to a simulated heatwave

Orlando Lam-Gordillo[1,2]*, Emily J. Douglas[1], Sarah F. Hailes[1], Andrew M. Lohrer[1]

**1** The New Zealand Institute for Earth Science, Hamilton, New Zealand, **2** College of Science and Engineering, Flinders University, Adelaide, Australia

* orlando.lam-gordillo@niwa.co.nz

## Abstract

Climate change is affecting ecosystems worldwide. Ocean temperature is increasing, leading to more frequent and intense extreme weather events such as heatwaves (HWs) and marine heatwaves (MHWs). These extreme temperature events can affect marine biodiversity, the relative success of different life history strategies, and the structure and function of whole ecological communities. Understanding the effects of HWs on marine communities and the mechanisms that enable marine communities to resist and recover from heat stress has become a research priority. Here, we investigated the effects of a simulated HW on macrobenthic communities and their functional traits in intertidal estuarine sediments using a novel in-situ seafloor warming experiment with two heatwave duration treatments. Our results revealed that the simulated HW influenced macrobenthic abundance and diversity, yet, functional traits and functional metrics were less affected. This suggests resilience to the simulated HW related to redundancy of macrobenthic functional traits. The idiosyncratic responses observed in our study reflects the complex and context-dependent macrobenthic responses to thermal extremes. Our findings suggest that maintaining functional trait diversity and redundancy could be key for boosting ecosystem resilience under increasing climatic extremes.

## Introduction

Global climate is warming due to the increasing release of anthropogenic greenhouse gases to the atmosphere [1–3]. The increasing global temperature is leading to hotter summers and more frequent extreme climatic events such as heatwaves [1,2,4,5]. In coastal ecosystems, marine heatwaves (MHWs) have already increased in intensity, frequency, and duration, with projections suggesting that this increasing pattern will continue in the future due to climate variability [2,4,6–8]. Intertidal coastal ecosystems are exposed to both marine and atmospheric heatwaves (HW) making them very vulnerable to climate change driven temperature anomalies [9].

**Data availability statement:** All relevant data are within the manuscript and its Supporting Information files.

**Funding:** This research was funded by the New Zealand Government's Strategic Science Investment Fund (SSIF) to the New Zealand Institute for Earth Sciences (ESNZ; FPRS2604 and FPRS2606).

**Competing interests:** The authors have declared that no competing interests exist.

Extreme temperature events including MHWs (i.e., prolonged periods – five or more days of sea surface temperatures above the 90th percentile) and HWs (i.e., prolonged periods of abnormally hot weather relative to the expected conditions) are likely to affect marine biodiversity, modifying the structure of communities, their distributional ranges, life history strategies, and survival [4,6,10–13]. These changes in biodiversity can have profound consequences to society, with losses of the essential ecosystem processes, functions, and services that these organisms and ecosystems provide to humankind [14–18]. Although several studies have assessed the effects of MHWs on biodiversity, focusing on kelp, macroalgae, coral reefs, and some marine invertebrates [e.g., 19, 20–23], less is known about the effects of HWs on intertidal macrobenthic assemblages and the factors affecting resistance and resilience to acute thermal stress in estuarine ecosystems.

Estuarine ecosystems are hotspots for biogeochemical cycling and biodiversity, and contribute to a wide range of ecosystem functions, services, and values (e.g., nutrient cycling, carbon storage, coastal protection, food production, cultural practices, amenity values) [24–26]. Estuarine macrobenthic fauna is generally well adapted to natural fluctuations such as tidal cycles and seasonal changes, particularly intertidal species that are exposed to air during low tide periods [27–29]. Estuarine macrobenthic organisms have the potential to cope with extreme weather events such as MHWs for short periods [30–32], however, negative impacts may occur when organisms face prolonged periods of elevated temperature that exceed their tolerance levels [31,33–35].

Marine heatwaves, alongside atmospheric heatwaves, can particularly affect the fitness of ectothermic estuarine organisms that cannot independently regulate their body temperatures and metabolic rates in a thermally fluctuating environment. These organisms are predicted to be more sensitive to temperature as global warming continues to increase, particularly when the temperature extremes are prolonged [36–39]. Because temperature fluctuations affect ectotherm metabolism [36,37,40], there may be associated consequences for growth and reproductive rates, maximum body sizes, feeding behaviours, bioturbation, and other vital functions of macrobenthos [31,35,37,41–43].

Understanding the mechanisms by which communities may resist and recover from HWs and MHWs has become a research priority. Some research has assessed the effects of HWs on intertidal rocky shores [e.g., 44], yet limited information is available for estuarine soft sediments. In soft-sediment systems, analysis of functional traits is expanding our understanding of organismal responses to thermal extremes and the mechanisms underpinning community and functional change. Functional traits are properties of organisms that can be measured (e.g., body size, life span, feeding mode, movement behaviours), usually at the individual organism level and used comparatively across species, facilitating the understanding of how different types of organisms present in a community respond to environmental change [45,46]. For example, mobile species or those capable of vertical migration within the sediment may avoid peak thermal exposure by relocating to cooler microhabitats, while deep-burrowing or tube-building taxa may experience buffered thermal regimes

compared to surface dwellers [31,32,46]. Functional diversity, defined as the collective combination and variety of functional traits expressed in an ecological assemblage that influence how an ecosystem operates or functions also plays a critical role in resilience [46–51]. Multiple metrics have been developed to quantify functional diversity, including functional richness, evenness, divergence, dispersion, and redundancy [50–55]. These complementary facets, all based on the analysis of functional traits and their modalities, provide complementary angles for understanding the relationships between biodiversity and how ecosystems function, i.e., the roles of organisms in communities and their links to ecosystem properties such as resistance and resilience to stressors [46,50–53]. Theory suggests that assemblages with high trait diversity are more likely to contain species that can maintain ecosystem functioning under stress, either through redundancy (i.e., multiple species sharing similar functional roles) or complementarity (i.e., different species compensating for different aspects of function) [51,56,57]. Identifying which traits boost resistance or facilitate recovery can improve predictions of ecosystem responses to climate extremes and inform conservation and management strategies.

In this study, we investigated the effects of a simulated heatwave during low tide on macrobenthic communities and their functional traits. We carried out an in-situ warming experiment with daily heat treatments applied for zero (control), five, or seven days. We aimed to (i) evaluate changes in macrobenthic communities, their functional traits, and functional diversity in different warming scenarios, (ii) investigate the relationship between thermal stress (simulated heatwave) and macrobenthic communities, functional traits, and functional diversity, and (iii) elucidate if the variability in responses of functional traits and functional metrics to thermal stress confers resilience to heatwaves. We hypothesised that macrobenthic taxonomical (i.e., abundance, richness, diversity) and functional metrics (i.e., single-trait indices (Community-weighted means), and multi-trait indices (Functional Richness, Evenness, Dispersion, Redundancy)) would decrease with a longer duration of warming (7 days, relative to 5 days) as temperature is a natural trigger of essential ecological processes with profound consequences if the thermal tolerance of the organisms is exceeded.

## Methods

This study was conducted as part of the same experimental framework described in Lam-Gordillo, Douglas (34) and Douglas, Lam-Gordillo (58), and followed identical field and analytical methodologies. Yet, the present work addresses a distinct research question, focusing specifically on understanding the effects of heatwaves on macrobenthic functional traits, which was not examined in the previous studies. In brief, we described the methods and experimental design used for this research below.

### Study area

The experiment was carried out in Waihī Estuary, in the Bay of Plenty region of the North Island of New Zealand (Fig 1). Waihī Estuary is a tidal lagoon type estuary (tidal range 1.7 m) that is permanently open to the sea and dominated by intertidal soft-sediment flats (57% of the estuary), which are fully submerged for several hours each tidal cycle [34,58]. The selection of the experimental site (Fig 1b-c) was based on a previous habitat mapping in the estuary [58] showing moderate muddy conditions (silt-clay: 26.65%), representing the overall condition of estuaries across New Zealand, and the site that previous studies had reported to be more influenced by simulated heatwaves (i.e., where there was a stronger experimental response signal) [34,59]

### Experimental design and set up

The experiment was conducted over a seven-day period in austral summer (17–23 February 2024). We deployed purpose-built Open Topped Chambers (OTCs) to the tidal flat in our experimental site (Fig 1d) at the start of each low tide period during seven days, following the protocols described by Lam-Gordillo, Douglas (34) and Douglas, Lam-Gordillo (58). In brief, the cone-shaped OTCs (80 cm base diameter, 30 cm top diameter) were constructed from transparent polycarbonate plastic (1.5 mm thickness) to promote the passive heating of enclosed sediments [34,59]. OTCs were deployed

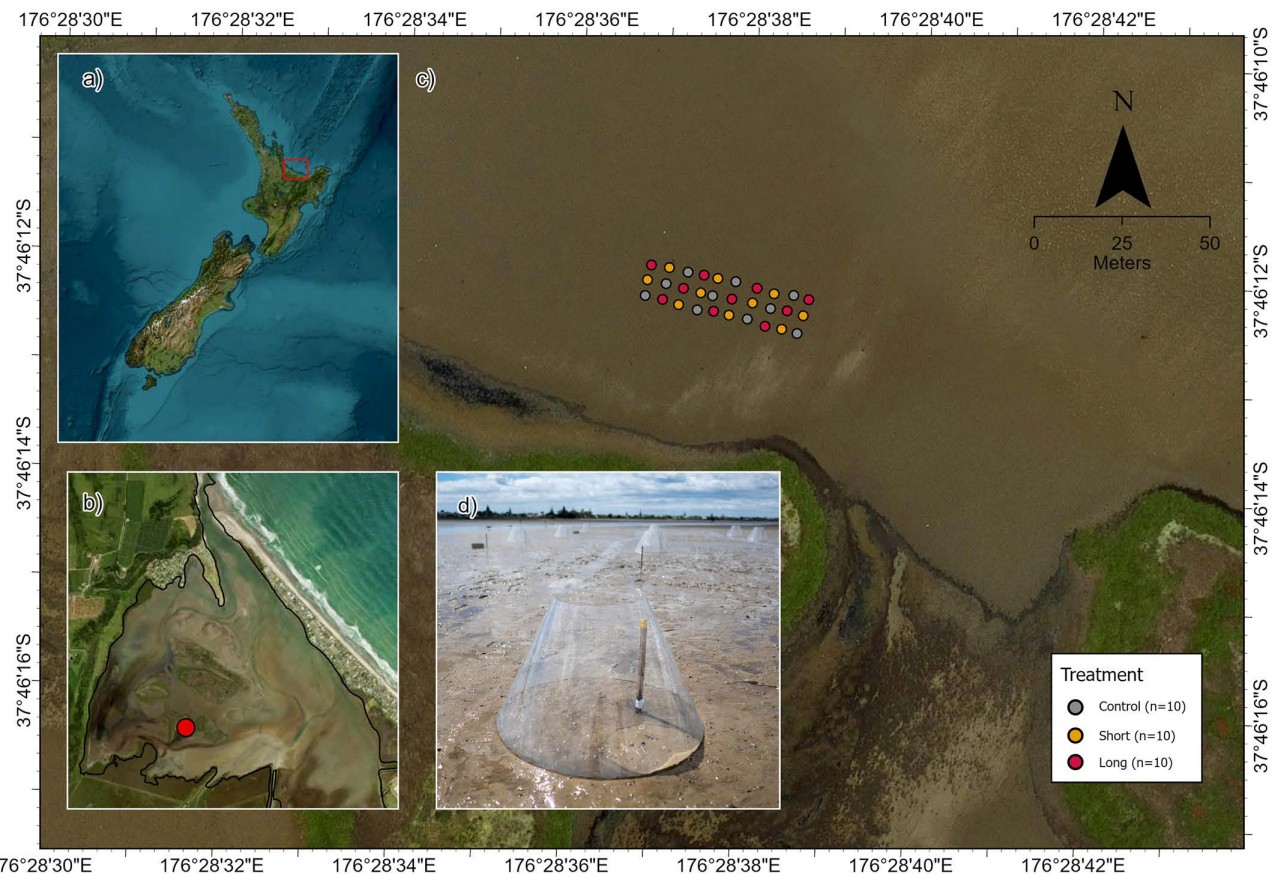

**Fig 1. Map of the study area showing (a) the location of Waihī Estuary in New Zealand, and (b) the area of Waihī Estuary where (c) the experimental plots were located.** (d) Close-up view of OTC showing position of stake with temperature loggers attached. Imagery retrieved from Land Information New Zealand (https://data.linz.govt.nz/).

(bottom edge pushed 1 cm into the sediment surface) for four hours during daytime low tides for a period five and seven days (Short and Long duration treatments, respectively) [34,59]. Experimental plots were unenclosed and OTCs were removed at night and during every high tide period during the experiment [34,59]. Ten replicate plots per treatment (Control: without OTCs, Short, Long; total n = 30) were established and arranged using a randomised block design which covered an area of 10 x 60 m with 5 m separation between each plot [34,59]. Our design enabled statistical analysis based on the factor "Treatment" (three levels: Control, Short, Long) [34,59].

## Data collection

**Temperature data.** Temperatures (°C) inside all control and treatment plots were measured using temperature loggers (EnvLoggers, Electric Blue CRL) located 1 cm below the sediment surface on wooden stakes (Fig 1d). Temperatures were recorded every minute and logged in all plots (Control, Long, Short) during the experiment. Temperature loggers remained in place throughout the experiment to ensure continuous temperature recording throughout the diurnal and tidal cycles.

## Macrobenthic fauna sampling

One core of sediment for the sampling of macrobenthos was collected from each of the control and experimental plots at the conclusion of each of the experimental treatment periods: Short treatment plots were sampled for macrofauna on

Day 5 (21/02/2024) and controls and long treatment plots were sampled on Day 7 (23/02/2024). Macrobenthic fauna were collected by hand using a PVC corer (13 cm diameter by 15 cm depth) near the centre of each plot. Immediately after collection, the entire volume of sediment collected in each core was sieved through a 0.5 mm mesh screen and preserved in 70% isopropanol. Benthic macrofauna were sorted, identified to the lowest possible taxonomic level following standardised protocols [60], and counted.

## Data analysis

Data recorded by the temperature loggers were analysed using the 'myClim' package [61] in R software [62]. Temperature data were first trimmed to only include times when the OTCs were deployed, and the resulting one-minute observations from each OTC period were averaged, providing one representative mean sediment temperature value per day per plot. In addition, we also calculated the minimum and maximum sediment temperature recorded during the times when the OTCs were deployed.

To test for differences in temperature between treatments, univariate PERMutational ANalysis Of VAriance (PERMANOVA) tests were conducted. PERMANOVA tests were based on Euclidean distance for the single variables (i.e., mean, minimum, and maximum temperature), permutation of residuals under a reduced model, Type III sums of squares, and 9999 permutations. In addition, multiple contrasts (pair-wise tests with 9999 permutations) were conducted if the fixed factor was significant in the main tests. Boxplots were constructed using the package 'ggpubr' [63] in R software [62].

To evaluate the effects of the simulated HWs on macrobenthic fauna, traditional and functional metrics were calculated using the packages 'vegan' [64] and 'FD' [54] in R software [62]. Traditional diversity metrics included taxonomic richness (S), Shannon-Wiener diversity (H', log e), and the total abundance (N) of macrofauna at each control and experimental plot. Functional metrics were calculated based on the selection of a suite of six functional traits and 27 traits modalities (Table 1; S1 Table). The functional traits selected describe behavioural, morphological, and physiological characteristics of the organisms that can be influenced by thermal stress. Trait information was retrieved from The New Zealand Trait Database for shallow-water marine benthic invertebrates – NZTD [65]. The NZTD uses a fuzzy coding procedure, assigning scores from 0 to 1 to each taxon, with 0 being no affinity and 1 being high affinity to a trait [for details see 65]. Four functional metrics were calculated including Functional Richness (FRic), Fucntional Evenness (FEve), Fucntional Dispersion (FDis), and Functional Redundancy (FR – calculated as the ratio between RaoQ entropy and H'). In addition, Community-level Weighted Means of trait values (CWM) were calculated to compare trait expression across treatments. Calculations of the four functional metrics were based on the combination of the macrobenthic taxa abundance matrix and the taxon-specific functional trait matrix using the R package FD [54]. The FD package produces an integrated data matrix by weighing the macrobenthic fauna abundances and functional traits data matrices, which is used for the calculation of all the functional metrics and CWM. To facilitate trend visualisation, taxonomic and functional metrics were presented as boxplots constructed using the package 'ggpubr' [63], while trends in CWM trait values are showed as circular plots created using the package 'ggplot2' [66]. To investigate the effects of simulated MHWs in metrics (abundance, S, H', FRic, FEve, FDis, FR, and CWM) between treatments, PERMANOVA was used to analyse variation among the Control, Short, and Long treatments, based on Euclidean distances for the single and multi macrobenthic metrics (i.e., abundance, S, H', FRic, FEve, FDis, FR, and CWM), permutation of residuals under a reduced model, sums of squares type III, and 9999 permutations [67]. In addition, multiple pair-wise tests were conducted to identify which treatment(s) contributed to the differences observed from PERMANOVA main tests.

To assess community and functional traits structure differences between treatments, bootstrapped non-Metric Multidimensional Scaling (nMDS) sample ordination plots were created based on a Bray-Curtis similarity resemblance applied to the macrobenthic abundance and functional trait data (i.e., the combination of macrobenthic abundance data and macrobenthic functional trait fuzzy coding data) [67]. To test for differences in community structure for both macrobenthic and trait data, PERMANOVA tests were performed using fourth root transformed macrofauna data, Bray Curtis Similarities, permutation of residuals under a reduced model, sums of squares type III and 9999 permutations [67]. Pairwise tests were also conducted if main effects were significant.

**Table 1. List of the functional traits and traits-modalities selected based on their response to extreme weather events [modified from 65]. Acronyms are used in Fig 4.**

| Functional trait | Trait modalities | Acronyms |
|---|---|---|
| Bioturbator | Biodiffusor | Bdiff |
| | Bioirrigator | Birrig |
| | No bioturbation | Nbio |
| | Surface modifier | Sumo |
| Body size | Large (20 mm) | Lar |
| | Medium (5–20 mm) | Med |
| | Small (0.5–5 mm) | Sma |
| Feeding mode | Deposit feeder | Defe |
| | Filter suspension | Fisus |
| | Grazer/scraper | Graz/Sc |
| | Predator | Pred |
| | Scavenger/opportunist | Scav |
| | Sub-surface deposit feeder | Ssdefe |
| Living habit | Attached | Att |
| | Burrower | Burr |
| | Free living/ Surface crawler | Free |
| | Parasite/ Commensal | Parc |
| | Tube dwelling | Tubdw |
| Movement method | Burrower | Burr2 |
| | Crawler | Craw |
| | None | Non |
| | Swimmer | Swim |
| Sediment position | Attached | Atta |
| | Bentho-pelagic | Be-pel |
| | Epibenthic | Epi |
| | Deeper >3 cm | Deep |
| | Surface shallow <3 cm | Surfsh |

Pearson correlations were performed to investigate the relationships between CWM traits, macrobenthic taxonomic and functional metrics and simulated MHWs conditions (mean, minimum, and maximum temperature) using the package 'corrplot' [68], and results were visualised through level plots constructed using the package 'lattice' [69].

## Results

### Simulated marine heatwave: intertidal sediment warming

Open Top Chambers increased the temperature of the sediments in the experimental plots compared to control plots successfully simulating a HW (Fig 2). Mean sediment temperature recorded in the experimental plots (i.e., Short and Long treatments) during the 7-day period ranged from 24°C to 34°C, while in the control plots ranged from 23°C to 28°C (Fig 2a). Mean sediment temperature was significantly different among treatments (PERMANOVA $p < 0.05$; Fig 2a). Pairwise post-hoc tests revealed that both Short and Long treatments were significantly warmer than Controls ($p < 0.05$; Fig 2a). Minimum temperature across the experimental plots ranged from 20°C to 30°C and in the control plots 20°C to 27°C, while the maximum temperature recorded ranged from 24°C and 36°C in the experimental plots and from 24°C to 30°C in the control plots (Fig 2a). Similar to mean sediment temperature, there were significant differences in minimum and maximum sediment

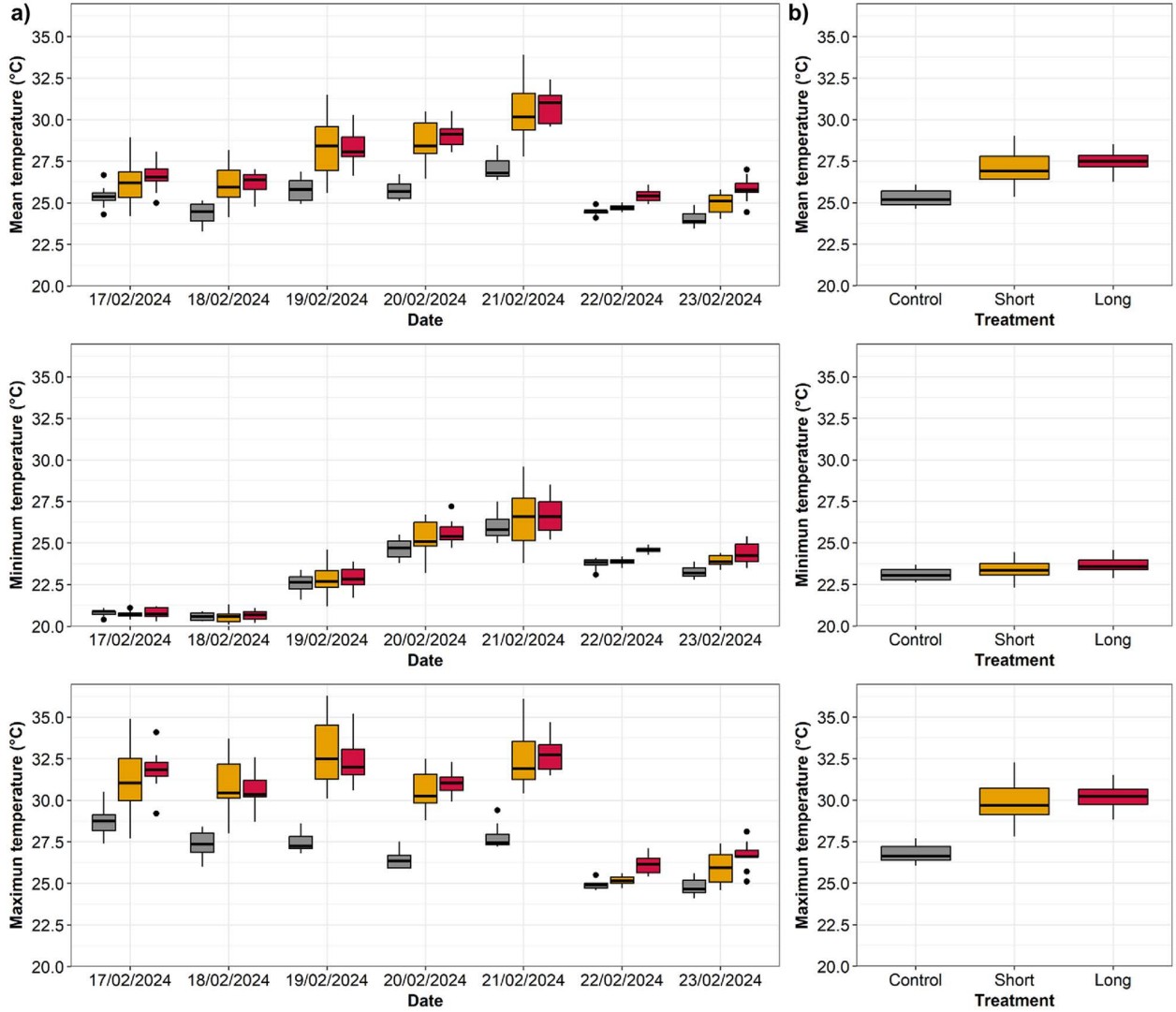

**Fig 2. Boxplots showing the sediment temperature (°C) recorded during the in-situ simulated heatwave experiment.** Column a) shows the mean, minimum, and maximum temperature recorded during each OTC period each day (seven consecutive days) for each treatment (Control, Short, Long, n = 10). Column b) shows the mean, minimum, and maximum temperature recorded during all OTC periods for the week (7 days) for each treatment (Control, Short, Long, n = 70). Note the lower temperatures and smaller control-treatment differences on days 6 and 7 due to the arrival of cooler, cloudier weather. Thick lines = median, dots = outliers.

temperature among treatments (p < 0.05; Fig 2a). Pairwise tests again showed that both Short and Long treatments were significantly warmer than Controls (p < 0.05; Fig 2a). The overall temperature (i.e., combining the temperatures from all the experimental days recorded during OTC incubations) showed similar patterns to the daily temperatures (Fig 2b). Mean, minimum, and maximum sediment temperature differed significantly between Controls and Long and Short treatments (PERMANOVA p < 0.05; Fig 2b), showing higher temperature in the experimental plots compared to the Controls.

**Effects of simulated heatwaves on macrobenthic and functional metrics**

In total, 8,769 macrobenthic organisms were collected and 25 macrobenthic taxa were identified in our experimental plots (S2, S3, and S4 Tables). Macrobenthic richness ranged from 9 to 26 taxa. The highest macrobenthic richness was

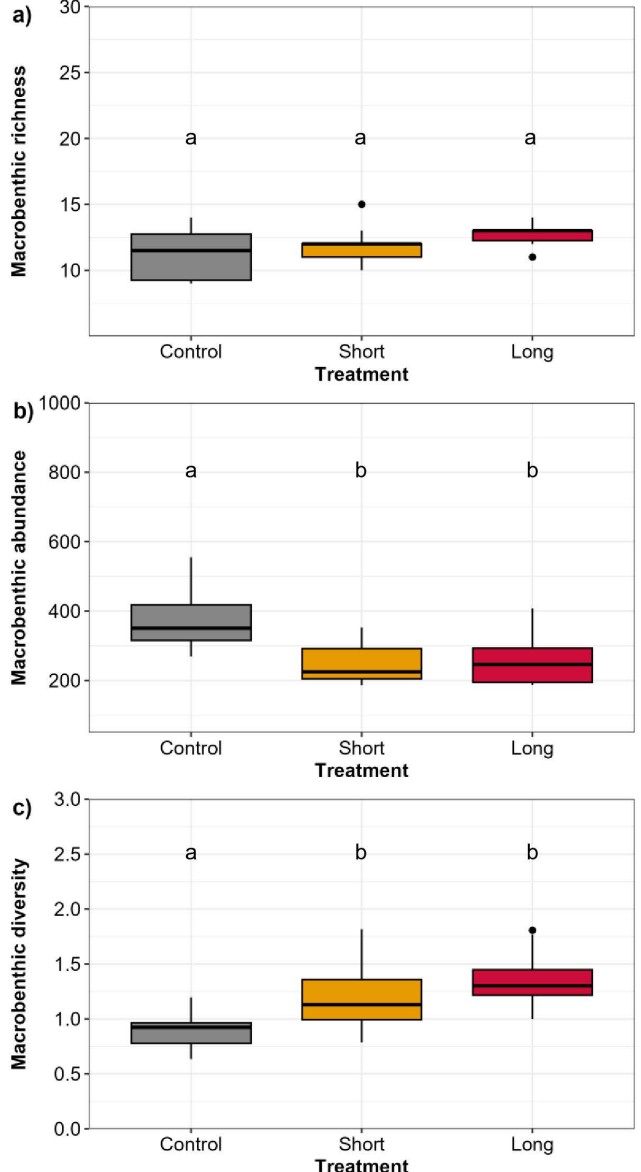

recorded in the Long treatment while the lowest richness was recorded in the Control treatment (Fig 3a). Median macrobenthic richness was higher in the Long treatment than the Control treatment, though there was no statistically significant difference across treatments (PERMANOVA p > 0.05; Fig 3a). Macrobenthic abundance ranged from 105 organisms per core in the Short treatment to 895 macrobenthic organisms per core recorded in Control treatment (Fig 3b). The highest macrobenthic median total abundance was recorded in the Control treatment, while the lowest macrobenthic median total abundance was found in the Short treatment (Fig 3b). Significant differences in macrobenthic abundance were found between treatments (PERMANOVA p < 0.05; Fig 3b), revealing significantly lower abundance in the Short and Long

**Fig 3. Boxplots showing the macrobenthic (a) richness (number of taxa), (b) abundance (total number of individuals per core⁻¹), and (c) diversity (H') across treatments (Control, Short, Long) in Waihī Estuary.** Thick lines = median, dots = outliers. Significant differences p < 0.05 between treatments are shown as "a, b".

treatments compared to the Control treatment (p < 0.05; Fig 3b). Macrobenthic diversity ranged from 0.63 to 2.38 (Fig 3c). The highest median macrobenthic diversity was found in the Long treatment and the lowest macrobenthic diversity in the Control treatment (Fig 3c). Significant differences in macrobenthic diversity across treatments were also found (PERMANOVA p < 0.05; Fig 3c), which followed the same pattern as macrobenthic abundance, showing significantly higher diversity in Short and Long treatments compared to Control treatment (p < 0.05; Fig 3c).

Across all treatments, the most expressed functional trait modalities, based on community-level weighted means (CWM) analyses of trait values, were surface modifiers (Bioturbation; contribution: 52%), medium body size (5–20 mm) (contribution: 100%), filter/suspension feeders (Feeding mode; contribution: 74%), tube dwelling (Living habit; contribution: 75%), burrowing (Movement method; contribution: 53%), and organisms positioned in shallow surface sediments (<3 cm; Living position; contribution: 50%) (Fig 4; S5 Table). Significant differences in functional traits and their modalities (as CWM) across treatments were identified (p < 0.05; Table 2; Fig 4; S5 Table). Most of the significant differences were found in the functional trait 'Feeding mode', for example, suspension feeders were significantly higher in the Control treatment compared to the Short and Long treatment, while predators and scavengers were significantly lower in the Control treatment compared to the Short and Long treatment (Table 2; Fig 4; S5 Table). Other significant differences in CWM trait modalities were identified in 'Bioturbation', 'Living habit', and 'Sediment position' and mostly between the Control and Long treatment (Table 2; Fig 4; S5 Table).

Significant differences in community and functional traits composition were detected between treatments (PERMANOVA p < 0.05; Fig 5). The bootstrapped nMDS plot for macrobenthic community structure revealed two main groups (Fig 5a). These groups were significantly different in community structure, differentiating the Control treatment from the Short and Long treatments (p < 0.05, Fig 5a). The macrobenthic taxa driving differences in community structure between Control and Short and Long treatments were the amphipod *Paracorophium excavatum* and the small bivalve *Arthritica sp.* Functional traits composition followed the same pattern as the community structure, revealing significant differences in functional trait structure between the Control treatment and the Short and Long treatments (p < 0.05; Fig 5b). The functional trait modalities that contributed the most to the differences between Control and Short and Long treatments were 'Streamlined" (Morphology), "Filter suspension" (Feeding mode), "Medium sized, 5-20 mm" (Body size), and "Tube dwelling" (Living habit).

Macrobenthic Functional Richness (FRic) ranged from 0.01 to 947.62 (S6 Table). FRIc was homogenous across treatments and no significant differences were found between treatments (PERMANOVA p > 0.05; Fig 6a). The highest median total FRic was recorded at the Long treatment, while the lowest median total was identified at the Short treatment (Fig 6a; S6 Table). Functional Evenness (FEve) ranged from 0.60 to 0.74 (S6 Table). The highest mean FEve was 0.55 recorded in the Short treatment, while the lowest mean FEve was 0.51 in the Long treatment (S6 Table). Significant differences in macrobenthic FEve between treatments were identified (PERMANOVA p < 0.05), and pairwise test revealed significant differences between the Short and Long treatments but no difference between Control and either Short or Long (p < 0.05; Fig 6b). Macrobenthic Functional Dispersion (FDis) ranged from 1.45 to 5.30 (S6 Table). FDis was significantly different across treatments (PERMANOVA p < 0.05) with significantly higher values in Short (mean value of 3.75) and Long (mean value of 4.03) treatments compared to the Control treatment (p < 0.05; Fig 6c; S6 Table). Macrobenthic Functional Redundancy (FR) ranged from 9.5 to 13.56. The higher mean FR value was in the Long treatment but was lower in the Short treatment (S6 Table). FR followed a similar pattern as FRic with no significant differences between treatments (PERMANOVA p > 0.05; Fig 6d; S6 Table).

Correlations between CWM trait modalities and the mean, minimum, and maximum temperature recorded during the in-situ warming experiment showed idiosyncratic positive and negative correlations (Fig 7). For example, in the Control treatment, significant positive correlations (p < 0.05) were found between the trait modality 'burrower' and the mean, minimum, and maximum temperature, relationships that were not evident in the OTC treatments. In the Short and Long treatments, the "bioirrigator", "scavenger", and "predator" trait modalities were positively correlated to temperature, compared

### a) Control

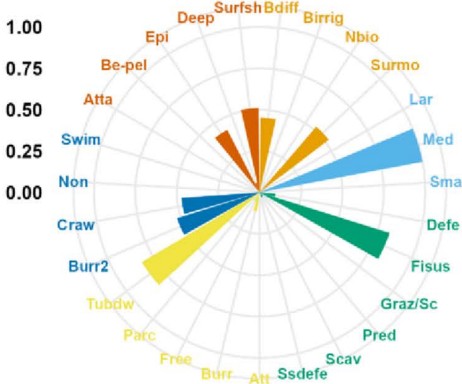

### b) Short

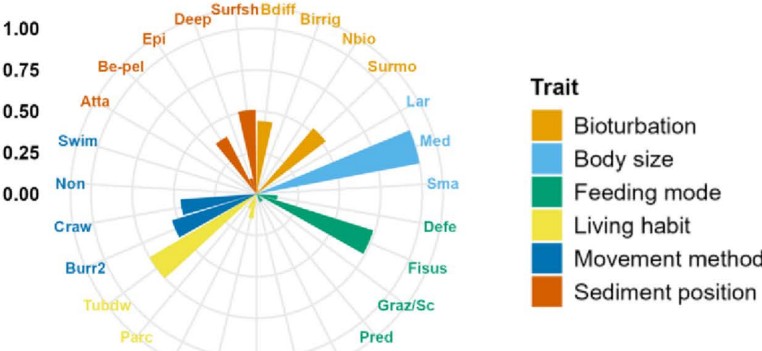

**Trait**
- 🟧 Bioturbation
- 🟦 Body size
- 🟩 Feeding mode
- 🟨 Living habit
- 🟦 Movement method
- 🟧 Sediment position

### c) Long

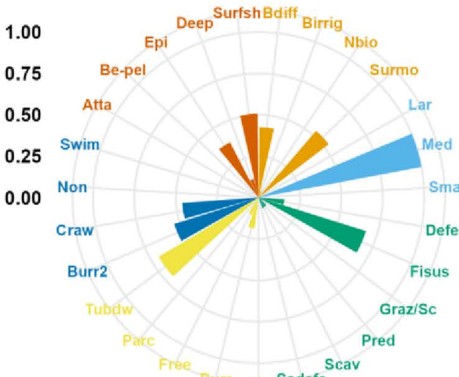

**Fig 4. Community-weighted means (CWM) of functional trait-modalities expression.** Scale represents the proportional contribution (each concentric line represents 0.25) to CWM recorded in (a) Control, (b) Short, and (c) Long treatments in Waihī Estuary. Trait modalities labels (acronyms) are defined in Table 1.

**Table 2. Test results from univariate one-way fixed factor PERMANOVA to compare functional trait modalities between treatments. Significant differences are shown in bold. NC: No computed.**

| Functional trait | Trait modality | Control vs Short | | Control vs Long | | Short vs Long | |
|---|---|---|---|---|---|---|---|
| | | t | P (perm) | t | P (perm) | t | P (perm) |
| Bioturbation | Biodiffusor | 0.6799 | 0.5058 | 1.9112 | 0.0777 | 1.1618 | 0.2644 |
| | Bioirrigator | 1.6882 | 0.1098 | 2.7380 | **0.0057** | 1.0558 | 0.3034 |
| | No bioturbation | NC | NC | NC | NC | NC | NC |
| | Surface modifier | 0.0430 | 0.9678 | 0.5249 | 0.6088 | 0.6538 | 0.5199 |
| Body size | Large (>20mm) | NC | NC | NC | NC | NC | NC |
| | Medium (5–20 mm) | NC | NC | NC | NC | NC | NC |
| | Small (0.5–5 mm) | NC | NC | NC | NC | NC | NC |
| Feeding mode | Deposit feeder | 1.7449 | 0.0972 | 2.6990 | **0.0100** | 1.1523 | 0.2677 |
| | Filter suspension | 2.4124 | **0.0212** | 4.0076 | **0.0002** | 1.3219 | 0.1970 |
| | Grazer/ Scraper | 0.6537 | 0.5251 | 2.3037 | **0.0360** | 1.9252 | 0.0726 |
| | Predator | 2.8357 | **0.0061** | 3.7977 | **0.0014** | 1.1233 | 0.2751 |
| | Scavenger/ Opportunist | 2.6013 | **0.0072** | 4.2949 | **0.0002** | 1.2788 | 0.2205 |
| | Sub surface deposit feeder | 1.1814 | 0.2736 | 1.2618 | 0.2311 | 0.0583 | 0.9469 |
| Living habit | Attached | NC | NC | NC | NC | NC | NC |
| | Burrower | 1.3723 | 0.1888 | 3.0606 | **0.0088** | 1.3242 | 0.2046 |
| | Free living/ Surface crawler | 2.2810 | **0.0298** | 3.8364 | **0.0017** | 0.7706 | 0.4463 |
| | Parasite/ Commensal | 0.9461 | 0.3707 | 1.0646 | 0.3224 | 0.0534 | 0.9662 |
| | Tube dwelling | 1.8337 | 0.0792 | 3.6780 | **0.0007** | 1.1752 | 0.2611 |
| Movement method | Burrower | 0.9731 | 0.3389 | 0.6805 | 0.5201 | 0.0579 | 0.9534 |
| | Crawler | 0.9346 | 0.3665 | 0.5573 | 0.5865 | 0.0993 | 0.9239 |
| | None | NC | NC | NC | NC | NC | NC |
| | Swimmer | 0.3760 | 1.0000 | 0.6236 | 1.0000 | 0.5566 | 0.5563 |
| Sediment position | Attached | NC | NC | NC | NC | NC | NC |
| | Bentho-pelagic | 0.5221 | 1.0000 | 0.6236 | 1.0000 | 0.2336 | 0.8609 |
| | Epibenthic | 2.4743 | **0.0246** | 2.6590 | **0.0114** | 0.6317 | 0.5348 |
| | Deeper than 3 cm | 2.8336 | **0.0100** | 3.3729 | **0.0018** | 0.8777 | 0.3849 |
| | Surface shallow <3 cm | 0.1455 | 0.8830 | 0.3285 | 0.7464 | 0.2677 | 0.8032 |

with weak or negative relationships in the Controls. In general, the long treatment showed more positive correlations compared to Control and Short treatments. All correlations, except for 'burrower' in the Control treatment, were not statistically significant (p > 0.05; Fig 7). Correlations between taxonomical and functional metrics and temperatures (i.e., mean, minimum, maximum) recorded in the in-situ experiment showed similar patterns across treatments (Fig 8). Significant negative relationships (p < 0.05) were identified between the FEve and mean, minimum, and maximum temperature in the Short treatment (Fig 8). Macrobenthic richness and abundance were positively correlated to temperature, similar to FRic and FDis. Except from FEve, whereas all other correlations were not statistically significant (p > 0.05).

## Discussion

Marine and atmospheric heatwaves are becoming more frequent, longer, and more intense globally. These extreme temperature events have repercussions when the thermal tolerances of species are exceeded, causing for example mass mortalities and shifts in community structure of marine assemblages [4,20,21,35,70,71]. In this study, we evaluated the effects of a simulated HW on macrobenthic communities and their functional traits using a novel in-situ intertidal warming experiment and two heatwave durations. Our findings revealed that the simulated HW influenced the macrobenthic fauna

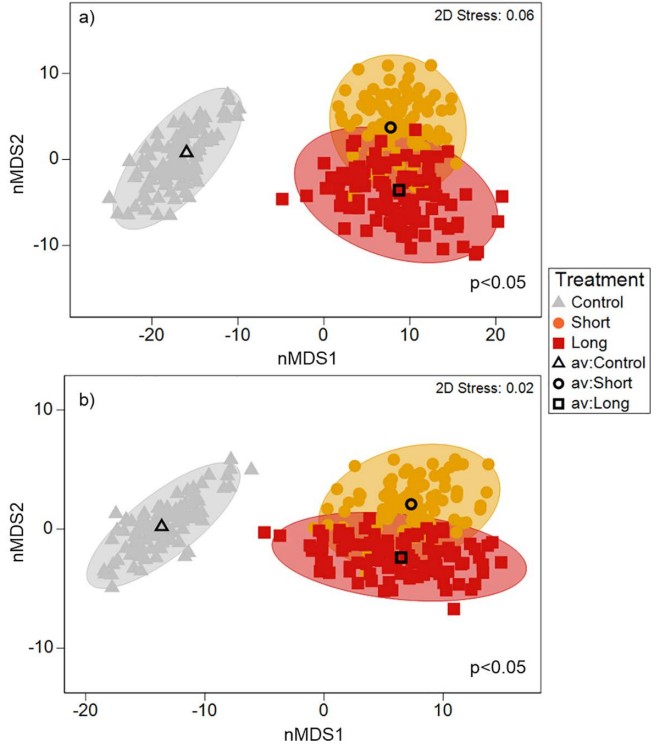

**Fig 5. Bootstrapped non-Metric Multidimensional Scaling (MDS) ordination plot showing the macrobenthic (a) community structure and (b) trait composition across treatments in Waihī Estuary.** Symbols close together in ordination space indicate greater similarity in multivariate composition.

taxonomical descriptors (i.e., abundance, diversity), however, functional traits and functional metrics were less affected, suggesting redundancy in functional diversity, which could enhance the macrobenthic resilience to short-term disturbances such as the simulated HW.

In contrast with our hypothesis, we found idiosyncratic responses to the simulated HW. Clear evidence of larger effects on macrobenthic fauna metrics (i.e., taxonomical and functional) with longer duration of warming was lacking. Our results demonstrated that the simulated HW influenced macrobenthic community descriptors, particularly macrobenthic abundance and diversity. These findings are consistent with previous studies reporting decreased abundance and altered species composition following natural or experimental heatwaves [35,71–73]. For example, in our study, most of the differences between the control and warming treatments were driven by decreases in the abundance of the amphipod *Paracorophium excavatum*. The mean abundance of this amphipod decreased by almost half in Short and Long (~90 and 108 organisms per core$^{-1}$, respectively) treatments compared to the abundance recorded in the Control (~183 organisms per core$^{-1}$). These effects could be attributed to direct physiological stress or increased metabolic demand leading to energy deficits, reproductive failure, or mortality [34,74–76]. However, again, consistent or stronger effects associated with longer heatwave duration were missing. This lack of clear duration-dependent effects suggests that the macrobenthic responses may be driven more by species-specific thermal tolerances or pre-existing environmental variability than by the heatwave duration alone [22,31,76,77]. Temporal mismatches between peak temperature exposure and sensitive life stages, or behavioural adaptations such as vertical burrowing in the sediment, could further modulate exposure and susceptibility to the simulated HW [31,32,78,79]. Diversity was significantly lower in Control treatment compared to Short and Long treatments, however, this effect was driven by the overall high abundance of dominant species (e.g., *Paracorophium*

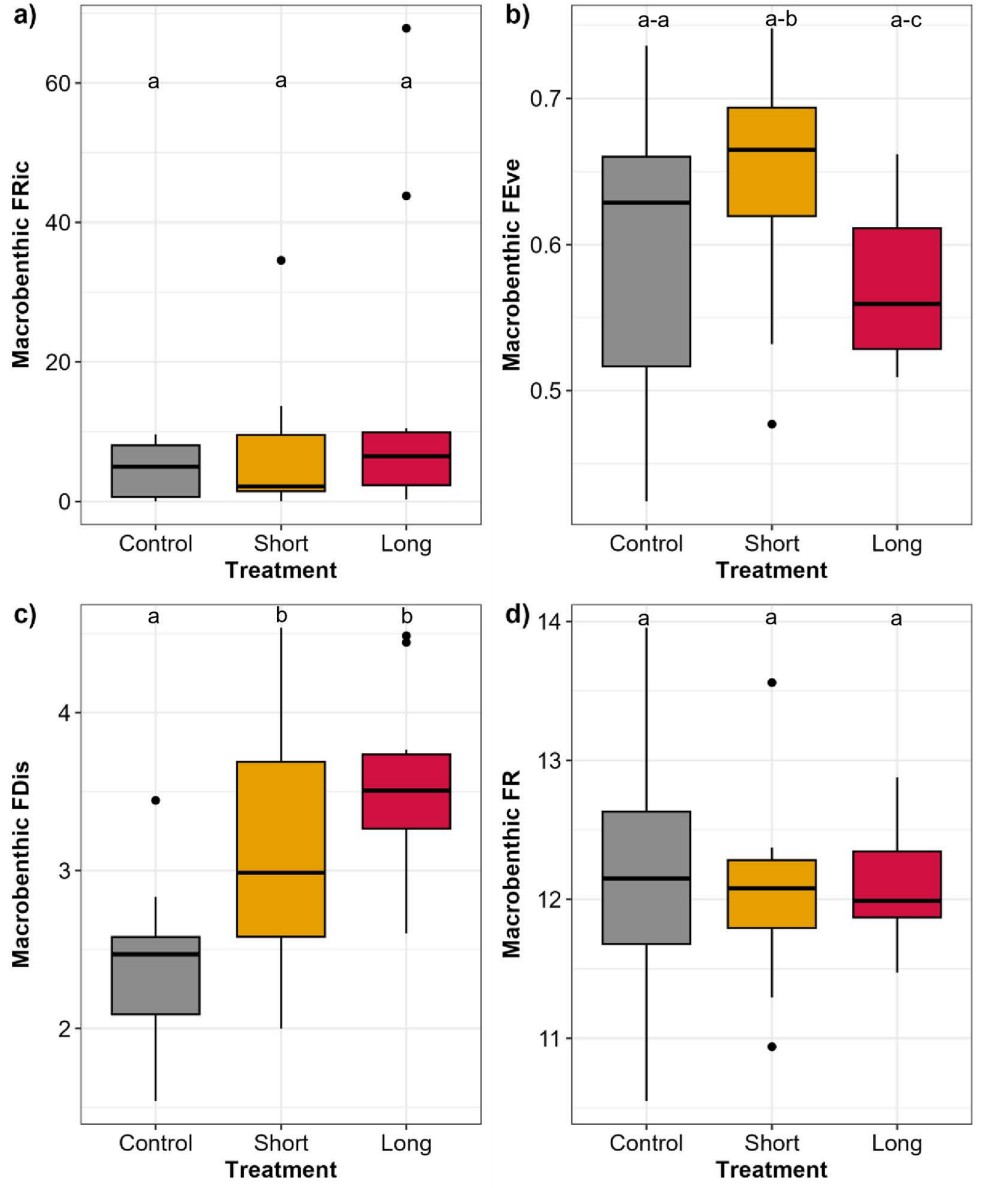

**Fig 6. Boxplots showing the macrobenthic (a) Functional Richness, (b) Functional Evenness, (c) Fucntional Dispersion, and (d) Functional Redundancy across treatments (Control, Short, Long) in Waihī Estuary.** Thick lines = median, dots = outliers. Significant differences p < 0.05 between treatments are shown as "a, b, c".

*excavatum*, oligochaete worms*, Arthritica sp.*), reducing the evenness in abundance across species and explaining the lower diversity in the Control treatment [80,81] rather than an effect due to the simulated HW. Furthermore, the sediment temperature measurements in our study were restricted to the upper 1 cm layer. This surface layer is ecologically relevant because many macrobenthic organisms occupy or interact with it during low tide, yet it does not capture potential thermal gradients deeper in the sediment [25]. Heat transfer to lower layers is expected to occur but at a reduced rate, and macro-benthic organisms capable of vertical migration may have experienced different thermal conditions than those recorded.

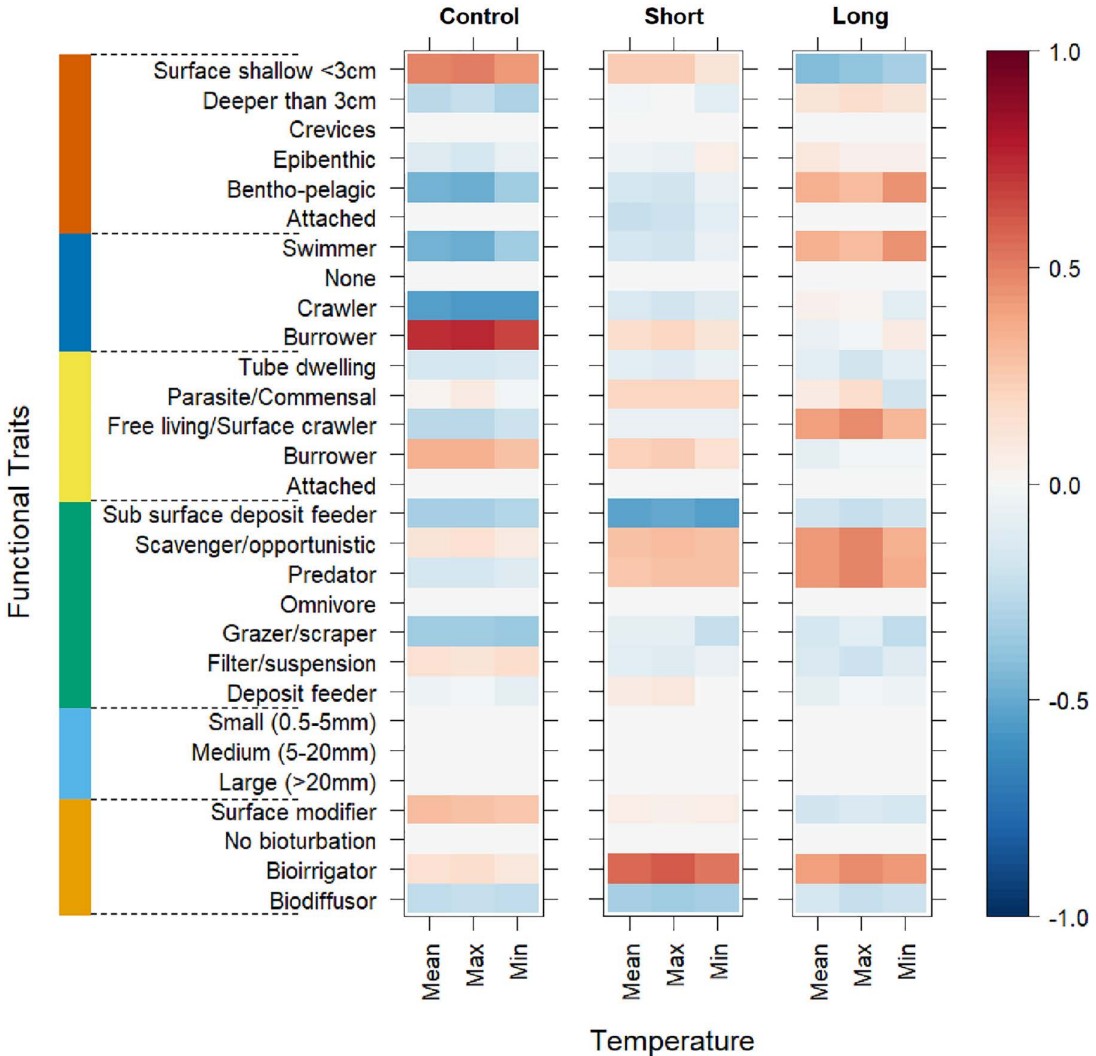

**Fig 7. Level plot showing the spearman correlations between the mean, maximum, and minimum sediment temperature and functional traits in each of the treatments (Control, Short, Long).** Functional traits – orange: Bioturbation, sky blue: Body size, green: Feeding mode, yellow: Living habit, blue: Movement method, vermilion: Sediment position. The colour scale indicates the magnitude of the correlations based on the Spearman correlation coefficient (rho ($\rho$) = + 1, 0, −1).

Differences in functional traits (as CWM) were found across treatments with most of the significant differences observed in the functional trait Feeding mode. For example, suspension feeders were significantly higher while predators and scavengers were significantly lower in the Control treatment compared to the Short and Long warming duration treatment. The elevated proportion of scavengers and the decreased proportion of filter/suspension feeders (as CWM) in the Short (5% and 73% respectively) and Long (6% and 67% respectively) treatments compared to Control (3% and 82% respectively) treatment likely reflect the cascading effects of thermal stress on benthic assemblages. Elevated temperatures can cause direct mortality or force organisms to emerge from the sediment, increasing their vulnerability to predation [31,32,71,35]. Such events create feeding hotspots that can attract mobile predators and scavengers from surrounding areas [70,82–84]. Similar scavenger aggregations have been documented following marine heatwaves and other disturbance events that elevate carrion availability on the seafloor, which may influence trophic interactions and nutrient cycling

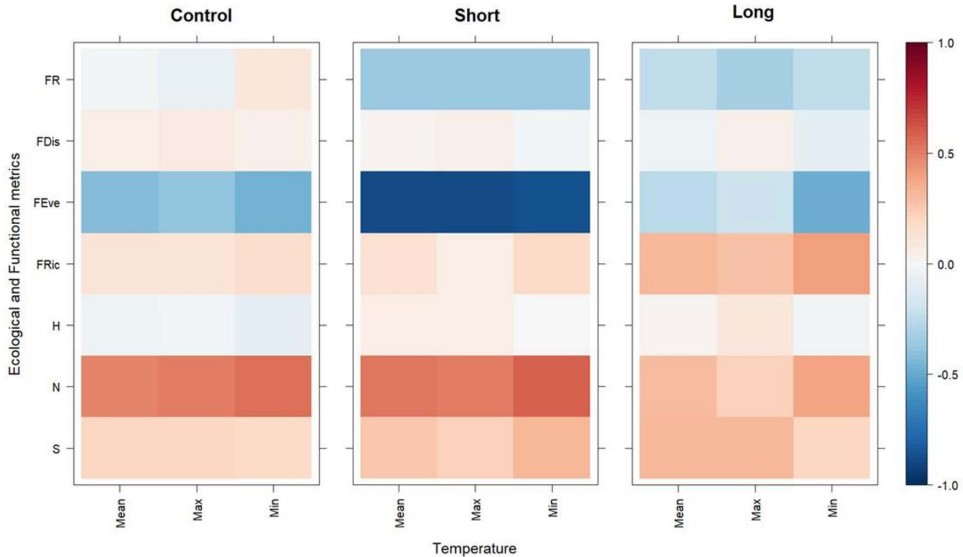

**Fig 8. Level plot showing the spearman correlations between the mean, maximum, and minimum temperatures recorded at each of the treatments (Control, Short, Long) and the taxonomical and functional metrics.** The colour scale indicates the magnitudes of the correlations based on the Spearman correlation coefficient (rho (ρ) = +1, 0, −1). S: richness, N: abundance, H: diversity, FRic: Functional Richness, FEve: Functional Evenness, FDis: Functional Dispersion, FR: Functional Redundancy.

[31,32,70,71,77,84,85]. Nevertheless, the influence of elevated temperatures on resource availability and trophic interactions, and the further potential mechanisms such as organisms surfacing or increased scavenging, were not directly tested or observed in our study and require further investigation. Functional Evenness in the Short and Long treatments was influenced by the simulated HW. Observed differences in FEve were high FEve in the Short treatment and low FEve in the Long treatment. These patterns could be related to the even distribution of trait strategies and efficient use of functional space but low redundancy in the Short treatment, in contrast to trait clustering and high redundancy but reduced functional niche breadth in the Long treatment [50–52]. However, the FEve recorded in the Control treatment was not significantly different to that in the Short and Long treatments.

In contrast to FEve, Functional Dispersion (FDis) was significatively influenced by the simulated HW. Higher FDis was recorded in the Short and Long treatments compared to the Control treatment. High FDis suggests that the functional traits are widely dispersed, the macrobenthic community exhibits broad functional niche breadth, high complementarity, and potentially higher ecosystem multifunctionality and resilience [50–53]. A previous study suggested [57] that disturbance often leads to a contraction of functional space and reduced FDis, yet, empirical responses are context-dependent such the findings in our study. In open, well-connected systems like an estuary, disturbance can promote colonisation by species with trait combinations not previously present, thereby expanding functional space and increasing FDis rather than reducing it [86,87]. For example, connectivity to adjacent habitats (seascape proximity or terrestrial–aquatic linkages) has been shown to maintain or even enhance functional complementarity by supplying species with complementary trait values [86,87]. Furthermore, the direction of change depends strongly on disturbance intensity, frequency and spatial scale, e.g., low-to-moderate disturbances or pulse disturbances can create opportunities for coexistence and trait overdispersion, whereas long-lasting, high-intensity disturbances more commonly lead to trait loss. Functional Richness and Functional Redundancy were less responsive to the simulated HW than Functional Evenness and Functional Dispersion. This observation suggests high redundancy of the macrobenthic

functional traits. Overall, the combined patterns from the functional metrics, i.e., Functional Richness, Evenness, Dispersion, and Redundancy, suggest a degree of resilience within the macrobenthic assemblages, even when community composition and abundance was altered. In other words, functional traits (e.g., "Bioturbator", "Feeding mode", "Movement method" and "Sediment position") and its modalities appear to remain well represented following heat exposure despite impacts on key individual species (i.e., *Paracorophium excavatum*, Oligochaeta, *Arthritica sp.*) that drove the composition structure differences between treatments.

Resilience based on functional traits could be explained by redundancy, where multiple taxa share similar trait modalities and therefore contribute similarly to ecosystem processes [22,50,51,57,88]. Our findings aligned with previous research suggesting weak or delayed effects of heatwaves on benthic ecosystem functioning [35,71,89]. Furthermore, species that differ taxonomically but share similar traits related to sediment mixing, feeding, or mobility may compensate for one another functionally. This redundancy could be acting as a buffer against thermal stress, maintaining processes such as bioturbation, bioirrigation, and nutrient cycling under changing environmental conditions [22,31,35,71,89,90]. It is also plausible that certain heatwave-resistant taxa (characterised by traits such as deep burrowing behaviour, deposit feeders, or broad thermal tolerances) persisted across treatments and contributed largely to the functional profile of the assemblage [22,31,91,92]. These traits may be acting as ecological filters, selecting for functionally important taxa under stress scenarios such as HWs.

The idiosyncratic nature of the responses observed in our study reflects the complex and context-dependent macrobenthic responses to thermal extremes. While temperature is a key driver of ectotherm organisms physiology, the response of macrobenthic communities is also shaped by multiple interacting environmental conditions and stressors such as anoxia, organic enrichment, eutrophication, and sediment characteristics [89,93–97]. These factors could potentially either amplify or dampen the effects of thermal stress, depending on local habitat characteristics. Moreover, the response of functional traits and functional metrics to heat stress may not manifest uniformly across time and space. Traits related to feeding mode, bioturbation, and reproductive strategy can be slow to change or exhibit lagged effects following a HW [22,28,29,98].

Our study supports the idea that maintaining and enhancing functional trait diversity and redundancy could be key for boosting ecosystem resilience under increasing climatic extremes. Protecting habitat heterogeneity and limiting cumulative anthropogenic pressures that affect macrobenthic communities, their functional traits and ecological functions, may help preserve ecosystem functioning in a rapidly warming world. It is important to highlight that the settings of our simulated heatwave experiment (e.g., only during low tide, ~4 hours of exposure per day, 5- and 7-day duration), represent a relatively subtle short-term thermal stress rather than prolonged or continuous warming, and therefore the patterns observed in macrobenthic metrics should be interpreted as short-term responses at the sediment surface. Longer or more intense heatwave events could produce different ecological outcomes, including greater shifts in trait composition or community structure. Future studies incorporating multi-depth temperature measurements and extended exposure durations (i.e., more days of heatwave) would help to assess whether the trends observed here persist or intensify under more severe conditions and provide a more comprehensive understanding of thermal stress effects on infaunal communities.

## Conclusion

We simulated an in-situ heatwave on estuarine intertidal sediments. Our results revealed variable responses in macrobenthic assemblages, with larger effects on macrobenthic abundance and diversity, while in general, functional trait responses remained relatively stable during the warming experiment. These findings suggest that benthic communities may exhibit a degree of short-term functional resilience to heat stress, likely mediated by the filtering and redundancy of functional traits. However, the absence of clear duration effects and the idiosyncratic nature of responses highlight the complexity of predicting community and functional dynamics under future climate extremes. Continued use of in-situ experimentation

and trait-based frameworks will be critical for improving our understanding of benthic ecosystem resilience in the face of climate change and increasing heatwaves.

## Supporting information

**S1 Table. Macrobenthic taxa and functional trait modalities assessed for the simulated heatwave in situ experiment.** Fuzzy coding was applied to each of the taxa ranging from 0 to 1.
(DOCX)

**S2 Table. Macrobenthic taxa list showing the taxa recorded in the simulated heatwave in-situ experiment.**
(DOCX)

**S3 Table. Macrobenthic abundance (organisms per core-1) collected in the control (C), short (S), and long (L) duration treatments from the simulated heatwave in situ experiment.**
(DOCX)

**S4 Table. Summary statistics of the macrobenthic taxa metrics recorded in simulated heatwave in situ experiment.**
(DOCX)

**S5 Table. Summary table of the macrobenthic CWM (Community-level Weighted Means) of trait values recorded in simulated heatwave in situ experiment.**
(DOCX)

**S6 Table. Summary statistics of the macrobenthic functional metrics recorded in simulated heatwave in situ experiment.**
(DOCX)

**S1 Dataset. Supplementary raw data. Macrobenthic abundance, diversity, and functional raw data and temperature raw data.**
(XLSX)

## Acknowledgments

We would like to thank Te Rūnanga o Ngāti Whakahemo and Professor Kura Paul-Burke (Waikato University) for inviting and warmly welcoming ESNZ team on to the Pukehina Marae and encouraging us to carry out the fieldwork and experimentation in Waihī estuary. We also thank the ESNZ Marine Ecology team in Hamilton, New Zealand for the processing of sediment and macrobenthic samples.

## Author contributions

**Conceptualization:** Orlando Lam-Gordillo, Emily J. Douglas, Andrew M. Lohrer.

**Data curation:** Orlando Lam-Gordillo, Emily J. Douglas, Sarah F. Hailes.

**Formal analysis:** Orlando Lam-Gordillo.

**Funding acquisition:** Andrew M. Lohrer.

**Investigation:** Orlando Lam-Gordillo, Emily J. Douglas, Sarah F. Hailes.

**Methodology:** Orlando Lam-Gordillo, Emily J. Douglas, Sarah F. Hailes, Andrew M. Lohrer.

**Validation:** Orlando Lam-Gordillo, Andrew M. Lohrer.

**Visualization:** Orlando Lam-Gordillo.

**Writing – original draft:** Orlando Lam-Gordillo.

**Writing – review & editing:** Orlando Lam-Gordillo, Emily J. Douglas, Sarah F. Hailes, Andrew M. Lohrer.

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
