## [Decision Letter · Decision Letter 0]

10 Nov 2025

Dear Dr. Lam-Gordillo,

Thank you for submitting your manuscript to PLOS ONE. After careful consideration, we feel that it has merit but does not fully meet PLOS ONE’s publication criteria as it currently stands. Therefore, we invite you to submit a revised version of the manuscript that addresses the points raised during the review process.

Dear Authors, two external reviewers have now assessed your manuscript "*Resilience of macrobenthic functional traits to simulated heatwave* ”, providing the comments that are reported below. As you can see, they both found your study interesting and generally worth of publication. At the same time, however, they identified a few issues that would require careful revision before this paper is recommendable for acceptance. Note that reviewer 2 had noticed some overlapping of this manuscript with other published and under review papers. Authors must pay special attention to this issue and provide clearly independent results that justify this paper.

Based on the reviewers' and my own assessment, I'm thus here inviting you to take all of these comments into careful consideration and to modify your manuscript according to the provided constructive suggestions. I will then be happy to receive and further examine your revised version together with a point-by-point reply to each comment by myself and each reviewer, where you will need to explain any changes done to a particular piece of text, or include supported and convincing counterarguments to any points you may disagree with I'm confident you will find the present comments and suggestions relevant and useful to improve your work and I'm thus looking forward to hearing back form you by the due time.

We look forward to receiving your revised manuscript.

Kind regards,

Marcos Rubal García, PhD

Academic Editor

PLOS ONE

https://www.nature.com/articles/s41598-025-86310-6?code=5e1d616b-c700-4549-a52d-9a13c81abaed&error=cookies_not_supported

In your revision ensure you cite all your sources (including your own works), and quote or rephrase any duplicated text outside the methods section. Further consideration is dependent on these concerns being addressed.

“This research was funded by the New Zealand Government's Strategic Science Investment Fund (SSIF) to the New Zealand Institute for Earth Sciences (ESNZ; FPRS2606).”

4. Thank you for stating the following in the Funding Section of your manuscript:

“This research was funded by the New Zealand Government's Strategic Science Investment Fund (SSIF) to the New Zealand Institute for Earth Sciences (ESNZ; FPRS2606).”

“This research was funded by the New Zealand Government's Strategic Science Investment Fund (SSIF) to the New Zealand Institute for Earth Sciences (ESNZ; FPRS2606).”

Additional Editor Comments (if provided):

Dear Authors, two external reviewers have now assessed your manuscript "Resilience of macrobenthic functional traits to simulated heatwave”, providing the comments that are reported below. As you can see, they both found your study interesting and generally worth of publication. At the same time, however, they identified a few issues that would require careful revision before this paper is recommendable for acceptance. Note that reviewer 2 had noticed some overlapping of this manuscript with other published and under review papers. Authors must pay special attention to this issue and provide clearly independent results that justify this paper.

Based on the reviewers' and my own assessment, I'm thus here inviting you to take all of these comments into careful consideration and to modify your manuscript according to the provided constructive suggestions. I will then be happy to receive and further examine your revised version together with a point-by-point reply to each comment by myself and each reviewer, where you will need to explain any changes done to a particular piece of text, or include supported and convincing counterarguments to any points you may disagree with I'm confident you will find the present comments and suggestions relevant and useful to improve your work and I'm thus looking forward to hearing back form you by the due time.

Reviewers' comments:

Reviewer's Responses to Questions

**Comments to the Author**

1. Is the manuscript technically sound, and do the data support the conclusions?

Reviewer #1: Partly

Reviewer #2: Partly

2. Has the statistical analysis been performed appropriately and rigorously?

Reviewer #1: Yes

Reviewer #2: Yes

3. Have the authors made all data underlying the findings in their manuscript fully available?

Reviewer #1: No

Reviewer #2: No

4. Is the manuscript presented in an intelligible fashion and written in standard English?

Reviewer #1: Yes

Reviewer #2: Yes

Reviewer #1: First, I would like to thank the authors for this interesting work, which I believe can make an excellent contribution to the study of functional diversity and the application of its metrics, which are necessary for a better understanding of how ecosystems function in the context of global climate change. Below is a general comment on the manuscript, followed by comments on each section.

Overall, the study is well conceptualised and structured, with a solid foundation in marine community ecology. The experimental concept is promising and provides valuable and interesting insights into how marine macrofaunal communities respond to the main effects of climate change, to which they are increasingly exposed. The experimental design is well constructed and balanced, supported by an adequate number of replicates, which gives the study good statistical representativeness and robustness.

However, in reporting and describing the study, several gaps were identified, particularly regarding sample collection, and an adequate description of the data processing for the calculation of the functional metrics, which are central to the study, was not provided.

As a result, it was not possible to consult the data (including means and standard deviations) obtained, as these were not clearly presented in either the main text or the supplementary materials. This makes it difficult to interpret the results, which are displayed only in the graphs, and also hinders comparison with other studies and a thorough evaluation of the work carried out.

For easier reading, I am also attaching a PDF file containing this information in this format, as well as a PDF file of the manuscript with the corresponding comments.

INTRODUCTION

Overall, the Introduction is well written, and the objectives of the study are clearly explained. However, it requires some additions. As the study mainly focuses on the “functional” aspects of the infaunal community, a brief, in-depth description and definition of functional diversity and its metrics is needed.

78: “Functional diversity also plays a critical role in resilience.” This is true, but it would be useful to spend some time explaining what functional diversity is and how it can be measured. I suggest briefly describing the metrics used to estimate functional diversity (which is not straightforward) and providing some references. Describing these metrics may help readers, including non-specialists, to better understand your results.

94-95: “functional metrics (i.e., individual functional traits, functional indices)”. It is not clear what type of indices you include under the definition of “individual functional traits”. In particular, it is not clear from the initial reading whether this refers to the organism level or to a single functional trait. I assume it refers to a single functional trait. If so, I suggest changing “individual functional trait” to “single-trait indices” and “functional indices” to “multi-trait indices”. “Single-trait indices” include, for example, the CWM that you mention later, while “multi-trait indices” include “functional richness, evenness”, etc. It would be more accurate to refer to these broader categories.

MATERIALS AND METHODS

The study and its objectives are promising, and it may provide new perspectives on understanding the responses of macrobenthic infaunal communities to more frequent events driven by global climate change. However, some aspects need clarification, from the experimental design and sample collection to data processing and analysis. Furthermore, for greater accuracy, it would be useful to include separate tables with the abundance matrix and species-trait matrix, perhaps in the supplementary materials. This would allow for a better understanding of the results.

Figure 1. Please check the caption and ensure that the letters correspond accurately to those indicated in the figure. Correct "e" to "d" in the figure.

111: Since you are discussing the OTCs, please refer directly to Figure 1d.

117-118: It is not clear whether the OTCs were removed during the night and at high tides. Please clarify.

119: For greater clarity, please specify that the “Control treatment” is without the OTCs.

129: Did you remove the temperature loggers from the control plots during the night and at high tide? How long is the tidal excursion? Did the logger measure temperature only in the top 1 cm of the sediment? If so, how can you be sure that the temperature also increases in the lower layers?

132-137: You mentioned the use of a PVC corer for macrofaunal sampling. How many cores did you collect per plot? Please specify. Furthermore, you used a PVC corer with a depth of 15 cm for macrofaunal sampling. Based on this, did you observe any sediment stratification? Did you consider all the macrofauna within the 15 cm, or did you collect it only from the surface layers? As you correctly mentioned in the introduction, some species may respond to thermal increases by moving within the sediment. Therefore, it would be interesting to observe this in the cores. Additionally, the procedure for macrofaunal sampling is not very clear. Please clarify: once collected, was the core rinsed and sieved through a 0.5 mm mesh?

160: Regarding Table 1, there is an inconsistency between the acronyms and the trait modalities for the traits “Living Habit”, “Movement Method”, and “Sediment Position”. Please check and correct.

165-169: It is not clear how you calculated the four metrics, particularly which data you used. Generally, obtaining these metrics requires a combination of the species abundance matrix and the trait matrix (with fuzzy coding). For greater clarity, please revise this section to specify how you treated the data before calculating the metrics. This clarification would help readers understand the results obtained. If you used a different procedure, please specify it. Also, indicate whether you performed standardisation or transformation of the data.

174: Are the factors mentioned related to the simulated heatwaves (5 days and 7 days)? This is not clear. Please specify the factors considered and the levels for each factor.

172-178: This section is somewhat difficult to follow and may cause confusion. PERMANOVA was used to test the effects of MHWs on taxonomic and functional metrics, correct? If so, it is redundant to restate the factors. If not, it is unclear which factors are being tested. As mentioned above, please specify the factors being analysed. Additionally, the phrase “the Euclidean distance for the single macrobenthic metrics” is unclear; it is not specified which metrics are being referred to. I recommend clarifying this short paragraph to avoid misunderstandings regarding the statistical analysis performed.

183: Do “Functional trait data” refer to the combination of the macrofaunal data and trait “fuzzy coding assignation”?

184-187: This section is clearer than the previous one. I suggest making changes to the previous section, using this one as a starting point.

RESULTS

Overall, the results are well presented and consistent with the Materials and methods section. However, I suggest further integration and more in-depth descriptions in the main text, as well as modifications to the plots. Generally, the choice of plot types is good, and they are easy to read, even for non-specialists. However, some plots need to be checked and adjusted to ensure consistency with the descriptions in the main text.

214-228: I suggest providing a supplementary table with the abundance and diversity data to support the interpretation of the boxplots. It may also be useful to describe in the main text the magnitude of these changes among the treatments (even if not significant in some cases), including values, as has already been done previously for temperature.

236-237: Please specify the p value.

247-252: How does the community structure change among the treatments? I suggest providing a description of the community structure in all treatments and indicating which species or taxa drive the arrangement in the nMDS. The same applies to the functional trait composition. It is also possible to overlay as vectors the species that contribute most to these dissimilarities. Please take this into consideration. Otherwise, it must be specified which main differences are supported by the statistical analysis.

254–265: As in the previous comment, for greater comprehensiveness, it may be useful to provide at least the mean values and standard deviations of the multi-trait indices (FRic, FEve, etc.) to better understand the magnitude of the differences. I suggest adding a supplementary table with the means of all the descriptors considered (abundance, richness, diversity, FRic, FEve, etc.) to support the reading of the main text and the plots.

270-272: Figure 7. In the caption, the sentence “The colour scale indicates the magnitude of the correlations” is assumed to refer to the colour gradient bar on the right. Does it represent the interval of the Spearman correlation coefficient (rho (ρ) = +1, 0, -1)? If so, please specify this in the caption and also next to the scale bar. Furthermore, if this is the case, there appears to be an inconsistency between the main text and Figure 7. According to the main text, a significant negative Spearman correlation was found between the Control and the trait modality “burrower”. In Figure 7, a negative correlation would be expected to be shown with a “blue” shade. Is the colour scale inverted? Please clarify this aspect to avoid misinterpretation of the results.

277-283: Figure 8. There is the same issue with the visualisation of the correlation. It appears that the interval is inverted: positive correlations are blue and negative correlations are red. Please ensure that the interval on the right matches exactly with the output of the level plot, as it currently appears inconsistent. As previously suggested, if the interval on the right (+1 / -1) corresponds to the Spearman coefficient (rho), please specify this next to the gradation scale in the plot and/or in the figure caption.

DISCUSSION

This section provides an overview of the study and its findings. However, although the arguments are valid and appropriate, they seem to enhance and reinforce the study and its findings only partially. The study required significant practical and interpretative effort, which I believe deserves greater recognition. While it is acknowledged that interpreting functional diversity metrics is not easy, if these metrics are used, it is necessary to provide a valid explanation of their selection and the type of information they convey. Only some of these metrics are explained, while others, which are also informative, are omitted. Furthermore, the arguments supporting ‘functional’ resilience are unconvincing because they rely solely on evidence of redundancy, which may result from bias in the selection of functional traits. I recommend that the authors supplement their argument by providing supporting data, for example in the form of tables. This would lend further weight to the arguments presented in this section and give due recognition to the work undertaken.

293-294: In my opinion, the findings related to the functional traits and metrics do not necessarily indicate resilience of the community but rather demonstrate redundancy in functional diversity during the experiment. Therefore, the sentence “however, functional traits and functional metrics were less affected, suggesting a degree of functional resilience” is problematic. Firstly, since Functional Evenness and Functional Dispersion changed significantly according to your results, this is inconsistent with your statement. Secondly, as I understand it, the experiment simulates a heatwave during low tide with relatively short exposure (4 hours per day, every day), alternating with high tide. Is this correct? Do you think this exposure is sufficient to expect changes in functional traits? Additionally, providing information on how many cores were collected from each plot each day would help to support this statement. Furthermore, a brief description of the community in the “Results” section could also help to predict the response to the heatwave experiment.

Furthermore, in selecting the functional trait, I notice some overlap between Living Habit and Movement Method, for example in the modalities “Crawler” and “Burrower”. The choice of trait depends on the structure of the community, and sometimes different species may share the same functional trait. Since trait selection is arbitrary and there is no standard rule for how many or which traits to consider, have you tried considering only one of the two functional traits in question—either “Movement Method” or “Living Habit”? This might reduce overlap and redundancy. How can you explain that the observed redundancy is a response of the community to the HWs, rather than a redundancy resulting from the “over-expression” of the same trait due to overlap between trait modalities?

296–298: Does the increase in temperature refer only to the first centimetre (0–1 cm) of the sediment? Did you observe thermal increases below the first sediment layer as well? If so, as you correctly mentioned in the introduction, some species might move within the sediment. In this case, documenting the effect of temperature increase could be challenging if the data refer only to the sediment surface.

330-346: What about the other indices, Functional Evenness and Functional Dispersion? According to your results, they change significantly between the control and treatments, as shown in the boxplot in Figure 6. It would be noteworthy to comment on their trends. Why does FDis increase significantly with exposure to the HW? The same question applies to FEve.

Reviewer #2: The authors conducted an in-situ warming experiment using open-top chambers (OTCs) in an intertidal estuary to assess heatwave-induced changes in macrofaunal community composition and functional trait resilience. The benthic community was sampled after 5 and 7 days of heating to additionally test the effect of increased warming duration. A key finding of this study is the persistence of community functions due to functional redundancy after a change in macrofaunal abundance and diversity.

The topic and overall objectives of this study are timely and relevant, and the results have the potential to contribute valuable insights into the resilience of intertidal benthic communities to heatwaves. However, the manuscript in its current form requires significant revision before it would be suitable for publication in a high-impact journal.

To my knowledge, this is the third manuscript arising from the same experimental setup, alongside the already published Lam-Gordillo et al., 2025 (“Effects of in situ experimental warming on metabolic expression in a soft-sediment bivalve”) and Douglas et al., in review (“Simulated heatwave alters intertidal estuary greenhouse gas fluxes: duration and degradation state determine response”). Given this context, it is essential that the present manuscript clearly articulates what new knowledge is being provided beyond those studies.

At present, there is substantial overlap in both methodology and findings. For example, the section “Simulated marine heatwave: intertidal sediment warming” closely replicates previously published results (e.g., “Seafloor temperature and sediment characteristics” in Lam-Gordillo et al., 2025. Additionally, a large part of the community composition results reiterates material already addressed in Douglas et al., in review. This leaves the functional-trait component as the primary novel contribution; however, as these analyses are limited to a single experimental site, the study does not fully leverage the broader context (muddy vs. sandy sites) that has been central to the complementary papers.

To strengthen the manuscript’s contribution, I strongly encourage the authors to (1) streamline or reference previously published overlapping results rather than repeating them, (2) clearly define the unique research question and interpretive framework that distinguishes this study from the others, and (3) consider expanding the functional-trait approach to include data from both experimental habitats where possible, as this would better reveal context-dependent responses and ecological relevance.

Language and format editing should also be revised throughout the manuscript.

More specific comments below

Title: Adjust the title to the scope of the study.

Introduction:

Paragraph 61-65: The previous paragraph states that intertidal macrobenthic communities are generally well adapted to extreme conditions, while this paragraph describes them as highly sensitive. Please clarify these contrasting statements.

Methods:

Line 125: “Environmental data” is a bit too general for just including temperature here.

Line 133: The methods mention that there are 10 plots for Control, Short, and Long treatments. Does this mean the Control plots were sampled only after 7 days? If so, could you clarify how comparable a 7-day Control is to the 5-day heating treatment, particularly given the changes in weather conditions during the final two days (Figure 2)? This has important implications for interpretation, as heating and exposure time are confounded and cannot be disentangled.

Results:

A detailed Section on “Simulated marine heatwave: intertidal sediment warming” is mostly unnecessary as Lam-Gordillo et al., 2025 already stated that the temperatures are significantly different between OTC and control. Furthermore, Douglas et al. (in review) confirmed significant difference and additionally specifies the actual temperatures. However, a shortened version for the context should remain. But these shortened information can also be presented in the methods section.

Throughout the results section: Please ensure accuracy when describing the boxplot figures. Boxplots display the median (not the mean), along with the interquartile range and potential outliers. Some statements currently interpret boxplots as showing mean values, which should be corrected for clarity and statistical accuracy (Lines 216-224, 256-260).

Line 238: The statement that “suspension feeders were significantly higher in the control treatment compared to the short and long treatment” seems difficult to evaluate as no quantitative data for suspension feeders are provided. Table 2 only reports test results, and Figure 4 shows proportional feeding mode contributions. Please include the underlying abundance data to support this conclusion.

Line 246: Please clarify if a nMDS or MDS was conducted.

Lines 256-265: This section exemplifies a need for caution when interpreting statistical comparisons among Control, Short, and Long treatments. As noted above, sampling time and heating duration are confounded, and therefore not all treatment contrasts represent clean tests of heating effects. Please clarify which comparisons are appropriate and adjust interpretations accordingly.

Lines 267-283: Positive and negative correlations described here do not match the colour scale presented in Figure 7. The colour scale shows dark red to be highly positively correlated and dark blue for highly negative correlation. Please clarify and realign your interpretations (or correct the figure).

Discussion:

Line 303-305: This is the first time where actual abundance data are presented. Given that community metrics are a central aspect of the study and can help explain several of the observed responses, it would be important to present these data more comprehensively elsewhere. In addition, when reporting abundance, please provide an appropriate unit (e.g., individuals per core, per square meter). Yet again the question – How reasonable is it to compare the Control abundance to Short treatment abundance?

Line 321-323: Repetition of results line 238.

Line 323-325: Where is the data showing the elevated densities of scavengers? Please also elaborate on the cascading effect and emerging of organisms and suggest more references on this discussion. Especially, as it was stated earlier (line 313) that burrowing into the sediment is a behavioural adaptation in response to heat stress. The suggested references, discussing stranded carrion and carrion cycling, are only vaguely applicable here.

Line 330: Please clarify “… were less responsive…” compared to?

Line 333-334: Be specific about which species. Furthermore, if the majority of the reduction in abundance is attributed to the three dominant species (as noted in line 316), it would be valuable to discuss their functional traits and roles in the community in order to place these changes into a broader context of community trait redundancy.

Cheers,

Norman Göbeler

**Do you want your identity to be public for this peer review?** For information about this choice, including consent withdrawal, please see our Privacy Policy

Reviewer #1: No

Reviewer #2: **Yes: ** Norman Göbeler

---

## [Author Response · Author response to Decision Letter 1]

8 Dec 2025

Ref.: Ms. No. PONE-D-25-44021

Resilience of macrobenthic functional traits to simulated heatwave

Dear Academic Editor Marcos Rubal García,

We thank you and the reviewers for constructive comments and the opportunity to amend the manuscript. In our revision, we have addressed all comments and suggestions from both reviewers.

RESPONSE TO THE EDITORIAL COMMENTS

Reply: Thank you for noting this. In this revised version of the manuscript, we have reviewed the PLOS ONE style requirements and ensured that the manuscript and all associated files now comply with the prescribed formatting guidelines. Track changes was disabled to avoid an overmarked effect across the manuscript.

2. We noticed you have some minor occurrence of overlapping text with the following previous publication(s), which needs to be addressed.

In your revision ensure you cite all your sources (including your own works), and quote or rephrase any duplicated text outside the methods section. Further consideration is dependent on these concerns being addressed.

Reply: Thank you for highlighting this concern. We have carefully reviewed the manuscript to identify and address any overlapping text with our previous publications. In the revised version, all duplicated content outside the Methods section has been rephrased or removed, and appropriate citations have been added for all sources, including our own prior work Lam-Gordillo et al. 2025 (Effects of in situ experimental warming on metabolic expression in a soft-sediment bivalve, Scientific Reports) and Douglas et al. 2025 (Simulated heatwave alters intertidal estuary greenhouse gas fluxes, Nature Communications).

“This research was funded by the New Zealand Government's Strategic Science Investment Fund (SSIF) to the New Zealand Institute for Earth Sciences (ESNZ; FPRS2606).”

Reply: Thank you for reviewing our financial statement. We have modified our financial disclosure to ensure PLOS ONE requirements are met, it reads as follow:

“This research was financially funded by the New Zealand Government's Strategic Science Investment Fund (SSIF) to the New Zealand Institute for Earth Sciences (ESNZ; FPRS2606, FPRS2604). The funders had no role in study design, data collection and analysis, decision to publish, or preparation of the manuscript”

4. Thank you for stating the following in the Funding Section of your manuscript:

“This research was funded by the New Zealand Government's Strategic Science Investment Fund (SSIF) to the New Zealand Institute for Earth Sciences (ESNZ; FPRS2606).”

“This research was funded by the New Zealand Government's Strategic Science Investment Fund (SSIF) to the New Zealand Institute for Earth Sciences (ESNZ; FPRS2606).”

Reply: As suggested by the editor, in this revised version of the manuscript, we have removed the funding statement from the manuscript, and included in this response letter:

“This research was financially funded by the New Zealand Government's Strategic Science Investment Fund (SSIF) to the New Zealand Institute for Earth Sciences (ESNZ; FPRS2606, FPRS2604). The funders had no role in study design, data collection and analysis, decision to publish, or preparation of the manuscript”.

5. We note that Figure 1 in your submission contain [map/satellite] images which may be copyrighted. All PLOS content is published under the Creative Commons Attribution License (CC BY 4.0), which means that the manuscript, images, and Supporting Information files will be freely available online, and any third party is permitted to access, download, copy, distribute, and use these materials in any way, even commercially, with proper attribution. For these reasons, we cannot publish previously copyrighted maps or satellite images created using proprietary data, such as Google software (Google Maps, Street View, and Earth).

Reply: Thank you for raising this point. Figure 1 uses imagery sourced from Land Information New Zealand (LINZ), which provides freely available geospatial data under an open license. These images are not proprietary and comply with the Creative Commons Attribution License (CC BY 4.0) requirements for PLOS ONE. We have included proper attribution in the figure caption and confirmed that the imagery is legally permissible for publication. Caption reads: “Fig 1. Map of the study area showing (a) the location of Waihī Estuary in New Zealand, and (b) the area of Waihī Estuary where (c) the experimental plots were located. (d) Close-up view of OTC showing position of stake with temperature loggers attached. Imagery retrieved from Land Information New Zealand (https://data.linz.govt.nz/).

6. Please include captions for your Supporting Information files at the end of your manuscript, and update any in-text citations to match accordingly.

Reply: As suggested by the editor, in this revised version of the manuscript we have included the captions for the Supporting information, lines 843-861.

Dear Authors, two external reviewers have now assessed your manuscript "Resilience of macrobenthic functional traits to simulated heatwave”, providing the comments that are reported below. As you can see, they both found your study interesting and generally worth of publication. At the same time, however, they identified a few issues that would require careful revision before this paper is recommendable for acceptance. Note that reviewer 2 had noticed some overlapping of this manuscript with other published and under review papers. Authors must pay special attention to this issue and provide clearly independent results that justify this paper.

Reply: Thank you for your feedback and for summarizing the reviewers’ assessments. We appreciate the recognition of the study’s relevance and the opportunity to improve the manuscript. In this revision, we have carefully addressed all reviewer comments and editorial suggestions. Specifically, we have ensured that the manuscript presents clearly independent results and interpretations that distinguish it from our previous related publications. Overlapping content has been minimized by rephrasing, removing redundant sections, and referencing previously published work where appropriate. Additionally, we have emphasized the novel contribution of this study, i.e., the effect of a simulated heatwave on macrobenthic functional traits, which was not covered in the other manuscripts. These changes aim to strengthen the originality and clarity of our manuscript.

Based on the reviewers' and my own assessment, I'm thus here inviting you to take all of these comments into careful consideration and to modify your manuscript according to the provided constructive suggestions. I will then be happy to receive and further examine your revised version together with a point-by-point reply to each comment by myself and each reviewer, where you will need to explain any changes done to a particular piece of text, or include supported and convincing counterarguments to any points you may disagree with I'm confident you will find the present comments and suggestions relevant and useful to improve your work and I'm thus looking forward to hearing back form you by the due time.

Reply: Thank you for your guidance and for summarizing the reviewers’ feedback. We appreciate the opportunity to improve the manuscript and have carefully considered all comments and suggestions. In this revision, we have provided a detailed point-by-point response to each comment from both reviewers and the editorial comments, explaining the changes made or providing reasoned clarifications where necessary. Our goal has been to ensure that the revised manuscript fully addresses the concerns raised and meets the standards for clarity, originality, and scientific rigor. We trust that these revisions strengthen the manuscript and look forward to your further assessment. Please see below the detailed reply to the reviewers’ comments.

RESPONSE TO THE REVIEWER’S COMMENTS

REVIEWER #1:

First, I would like to thank the authors for this interesting work, which I believe can make an excellent contribution to the study of functional diversity and the application of its metrics, which are necessary for a better understanding of how ecosystems function in the context of global climate change. Below is a general comment on the manuscript, followed by comments on each section.

Reply: Thank you for your positive assessment of our work and for recognizing its potential contribution to the study of functional diversity and ecosystem functioning under climate change. We appreciate your constructive feedback and have carefully considered all subsequent comments to improve our manuscript. In the revised version, we have addressed each point in detail and provided explanations or clarifications where necessary to ensure the manuscript meets the highest standards.

Overall, the study is well conceptualised and structured, with a solid foundation in marine community ecology. The experimental concept is promising and provides valuable and interesting insights into how marine macrofaunal communities respond to the main effects of climate change, to which they are increasingly exposed. The experimental design is well constructed and balanced, supported by an adequate number of replicates, which gives the study good statistical representativeness and robustness.

However, in reporting and describing the study, several gaps were identified, particularly regarding sample collection, and an adequate description of the data processing for the calculation of the functional metrics, which are central to the study, was not provided.

Reply: Thank you for your positive evaluation of the study’s conceptualization and design. We appreciate your recognition of the robustness of the experimental approach and its relevance to understanding climate change impacts on marine macrofaunal communities. Your feedback reinforces the importance of this work, and we have taken care to maintain the clarity and rigor of the manuscript while addressing all subsequent comments to further strengthen its contribution.

In the revised manuscript, we have added detailed descriptions of the sample collection procedures and clarified the steps involved in data processing for the calculation of functional metrics in the Methods section. Specifically, we have now explained how the species abundance matrix and trait matrix were prepared, including fuzzy coding, standardization, and the methods used to compute single-trait and multi-trait indices, in addition of presented the macrobenthic abundance raw data in the supplementary materials. These clarifications aim to improve transparency and allow readers to fully understand the analytical workflow.

As a result, it was not possible to consult the data (including means and standard deviations) obtained, as these were not clearly presented in either the main text or the supplementary materials. This makes it difficult to interpret the results, which are displayed only in the graphs, and also hinders comparison with other studies and a thorough evaluation of the work carried out.

Reply: Thank you for highlighting this issue. In the revised manuscript, we have included the macrobenthic abundance raw data in the supplementary materials as a new table to ensure transparency and facilitate comparison with other studies. Additionally, we have referenced these values in the main text where relevant to support interpretation of the graphical results. These additions aim to improve clarity and allow for a more thorough evaluation of the work.

For easier reading, I am also attaching a PDF file containing this information in this format, as well as a PDF file of the manuscript with the corresponding comments.

INTRODUCTION

Overall, the Introduction is well written, and the objectives of the study are clearly explained. However, it requires some additions. As the study mainly focuses on the “functional” aspects of the infaunal community, a brief, in-depth description and definition of functional diversity and its metrics is needed.

Reply: Thank you for the positive comments. As suggested by the reviewer we have provided the definition of functional diversity and related metrics in the introduction section. Lines 82, and 88-96.

78: “Functional diversity also plays a critical role in resilience.” This is true, but it would be useful to spend some time explaining what functional diversity is and how it can be measured. I suggest briefly describing the metrics used to estimate functional diversity (which is not straightforward) and providing some references. Describing these metrics may help readers, including non-specialists, to better understand your results.

Reply: As suggested by the reviewer, in this revised version of the manuscript we have provided the definition of functional diversity and further explain how it is calculated. We also have detailed the metrics related to functional diversity and provide the relevant references in the Introduction section, lines 88-96, and provided extended clarification on the calculations and metrics used in the Methods section, lines 206-208.

94-95: “functional metrics (i.e., individual functional traits, functional indices)”. It is not clear what type of indices you include under the definition of “individual functional traits”. In particular, it is not clear from the initial reading whether this refers to the organism level or to a single functional trait. I assume it refers to a single functional trait. If so, I suggest changing “individual functional trait” to “single-trait indices” and “functional indices” to “multi-trait indices”. “Single-trait indices” include, for example, the CWM that you mention later, while “multi-trait indices” include “functional richness, evenness”, etc. It would be more accurate to refer to these broader categories.

Reply: As suggested by the reviewer, in this revised version of the manuscript we have modified the wording and adopted the terms “single-trait indices and multi-trait indices”. Lines 114-116, “… and functional metrics (i.e., single-trait indices (Community-weighted means), and multi-trait indices (Functional Richness, Evenness, Dispersion, Redundancy))…”.

MATERIALS AND METHODS

The study and its objectives are promising, and it may provide new perspectives on understanding the responses of macrobenthic infaunal communities to more frequent events driven by global climate change. However, some aspects need clarification, from the experimental design and sample collection to data processing and analysis. Furthermore, for greater accuracy, it would be useful to include separate tables with the abundance matrix and species-trait matrix, perhaps in the supplementary materials. This would allow for a better understanding of

---

## [Decision Letter · Decision Letter 1]

17 Dec 2025

Dear Dr. Lam-Gordillo,

Thank you for submitting your manuscript to PLOS ONE. After careful consideration, we feel that it has merit but does not fully meet PLOS ONE’s publication criteria as it currently stands. Therefore, we invite you to submit a revised version of the manuscript that addresses the points raised during the review process.

Dear Authors,

I have received the reports from referees on your manuscript, "*Redundancy of macrobenthic functional traits boosts resilience to a simulated heatwave* ", submitted to Plos One.

Based on the advice received, I have decided that your manuscript will be recommended for publication after you have carried out the final suggestions by referees.

Best regards

We look forward to receiving your revised manuscript.

Kind regards,

Marcos Rubal García, PhD

Academic Editor

PLOS One

Journal Requirements:

Reviewers' comments:

Reviewer's Responses to Questions

**Comments to the Author**

Reviewer #1: (No Response)

Reviewer #2: (No Response)

2. Is the manuscript technically sound, and do the data support the conclusions?

Reviewer #1: Yes

Reviewer #2: Partly

3. Has the statistical analysis been performed appropriately and rigorously?

Reviewer #1: Yes

Reviewer #2: Yes

4. Have the authors made all data underlying the findings in their manuscript fully available?

Reviewer #1: Yes

Reviewer #2: Yes

5. Is the manuscript presented in an intelligible fashion and written in standard English?

Reviewer #1: Yes

Reviewer #2: Yes

Reviewer #1: I would like to thank the authors for carefully revising this manuscript, which I still believe may be an excellent contribution to the study and application of functional metrics, necessary to understand and predict how ecosystems may respond to the effects of global climate change.

Overall, the authors have thoroughly revised the manuscript by checking and correcting inconsistencies between the main text and the graphs. They have also properly integrated the manuscript with the required information and explanations regarding the experimental design and methodologies used, as requested. Furthermore, they have provided sound and valid counterarguments to the comments expressed in the introduction and discussion sections, and have rightly added supplementary information, allowing for a better elaboration of the data obtained and a clearer interpretation of the results.

In light of the comments and revisions made, I support the acceptance of the manuscript. However, I ask the authors to further check and clarify a few aspects that have emerged after a careful assessment of the data provided in the supplementary materials, which were not previously presented.

I consider this further minor revision a crucial step towards ensuring the scientific validity and consistency of the study, and ultimately necessary for acceptance.

Below, you can find some additional comments concerning the “Data analysis” section and the “Results”.

For easier reading, I am also attaching a PDF file containing this information in this format.

Materials and Methods

Data analysis

Line 188 and S1 Table: Firstly, thank you for providing this table. The authors mentioned that no data standardisation or normalisation was performed. Clarification of the values presented in the tables is required. Is this combined matrix obtained by merging species abundances with functional categories? Such merging would provide more reliable quantitative information regarding the functional aspects of the community. Otherwise, this table appears to provide only qualitative information related to the presence of each species or taxon and their relative affinity for certain trait modalities, without weighting by the number of individuals (i.e. abundance) found. Consequently, the values are likely to be low in dispersion, and there may be no need for standardisation or normalisation. In my opinion, a quantitative trait analysis would be more informative in the context of this study, and I would suggest considering this type of analysis, as it could add robustness to the work. I would like to clarify that the procedure performed by the authors is correct, but it may be less informative and contextualised, since variations in species abundances are primarily indicative of changes in community structure when exposed to disturbance. If the authors wish to consider my suggestion to obtain a new merged species-trait matrix weighted by abundance, it is necessary to ensure consistency in the calculation of other functional metrics, including CWM. In this case, standardisation or normalisation may also be required. For example, the high abundance of the amphipod Paracorophium excavatum may not be comparable with the lower abundances of other species (e.g. Paradoneis lyra).

Lines 205-209: It is stated that “Euclidean distance” was used for both structural metrics (i.e. abundances, S, and H’) and functional metrics (FRic, FEve, FDis, FR, and CWM). However, as the functional trait analysis includes both categorical traits (e.g. feeding mode, movement method, etc.) and continuous traits (e.g. body size), the literature recommends using the Gower dissimilarity index (Gower, 1971; Teichert et al., 2017). Euclidean distance is typically applied only to continuous (environmental) variables.

Results

Lines 299-301: The FRic values are surprisingly high given the number of species found (25). A high FRic indicates a very wide trait space, and this index is usually closely linked to species richness.

Lines 304-305: Please verify that the FEve values exactly match those provided in Table S6. (FEve values range from 0 to 1.)

Reviewer #2: Review of 1st revision of Lam-Gordillo et al. “Redundancy of microbenthic functional traits boosts resilience to a simulated heatwave” (PLOS One, December 2025)

The manuscript has experienced a thorough revision after its initial submission and the authors have addressed all remarks and I particularly appreciate the effort invested into the data provision and strengthening the ecological context of the study. Most of the issues have been sufficiently resolved, but a few comments remain. The referred line numbers below are according to the track change document.

Lines 61: Please clarify these contrasting statements. Line 61: “Estuarine macrobenthic organisms are likely well adapted to cope with extreme weather events such as MHWs …” and line 69: “… these organisms are predicted to be more sensitive to temperature extremes”

Lines 165 (Macrobenthic fauna sampling): Please add the sampling dates for the macrofauna. The current description is confusing due to inconsistencies between the manuscript and the revision responses. Specifically, the manuscript describes experimental treatment days, while the responses alternatively state that (i) Control and Long (7-day exposure) were sampled after 7 days, with the Short (5-day exposure) treatment sampled after 5 days, and (ii) that all treatments were sampled simultaneously after the 7-day experimental period. Please clarify.

Lines 242–249 and Fig. 2: Please clarify whether temperature differences between the Short and Long treatments were statistically tested. In addition, it is difficult to identify from Fig. 2 when the OTC chambers were deployed during the Short treatment, as temperatures appear consistently higher than the Control across all days. If the Short treatment was sampled at the same time as the other treatments (23/02/2024?) – Would you have an explanation why the temperature was elevated on 17 and 18/02/2024?

Lines 394–403: While the example of functional traits (e.g. suspension feeders, predators, scavengers) in the Results section (lines 287–291) is appropriate, caution is needed when interpreting and linking these traits, as the discussion establishes an apparent link between them that is not directly supported by the data. To my knowledge, a large body of literature indicates that a common response of benthic invertebrates to excessive heat is deeper burrowing into the sediment (agreeing with your suggestion in line 377), rather than surfacing, unless driven by hypoxia, which I don’t suppose to be the case here. The manuscript suggests surfacing of animals and links this to a trophic cascade of increased resource availability for scavengers feeding on carrion and nutrient cycling. Given the lack of direct observations or data on animal movement in this study, such an interpretation is not sufficiently supported and should be reconsidered or more cautiously framed. In addition, given the availability of data now, I ask the authors to re-examine Table S5. The reported trait proportions (3% in Control, 5% in Short, and 6.8% in Long) appear to correspond to predators rather than scavengers (0.4%, 0.8%, 0.8%, respectively). Please verify this point and revise the text and discussion accordingly.

Line 413: According to Mouillot et al. (2013, “A functional approach reveals community responses to disturbances”), FDis is generally expected to decrease under disturbance due to the loss of functions. In contrast, in this study, FDis significantly increases. The authors may wish to contextualize these results – Could the observed pattern be related to the openness of the study system?

Recommended (constructive, clearly optional)

Given that the authors have conducted extensive analyses in related work, resulting in two previously published articles, it would be valuable to further interpret the observed changes in functional traits in relation to the metabolic responses of Austrovenus stutchburyi and/or, potentially, to changes in greenhouse gas fluxes. For example, the significantly increased proportion of bioirrigators in the Long treatment (and the near-significant increase in the Short treatment) could be discussed in the context of the elevated CO₂ and CH₄ influxes reported in Douglas et al. (2025). This additional interpretation is optional and, although speculative, I find it highly interesting. It could further enhance the ecological integration of the study.

**Do you want your identity to be public for this peer review?** For information about this choice, including consent withdrawal, please see our Privacy Policy

Reviewer #1: No

Reviewer #2: **Yes: ** Norman Göbeler

---

## [Author Response · Author response to Decision Letter 2]

18 Dec 2025

Ref.: Ms. No. PONE-D-25-44021R1

Redundancy of macrobenthic functional traits boosts resilience to a simulated heatwave

Dear Academic Editor Marcos Rubal García,

We thank you and the reviewers for constructive comments and the opportunity to revise our manuscript. In this new revision, we have addressed all comments and suggestions from both reviewers.

RESPONSE TO THE REVIEWER’S COMMENTS

REVIEWER #1:

I would like to thank the authors for carefully revising this manuscript, which I still believe may be an excellent contribution to the study and application of functional metrics, necessary to understand and predict how ecosystems may respond to the effects of global climate change.

Reply: Thank you for your positive assessment of our manuscript and for recognizing its potential contribution to advancing the study and application of functional metrics in predicting ecosystem responses to climate change.

Overall, the authors have thoroughly revised the manuscript by checking and correcting inconsistencies between the main text and the graphs. They have also properly integrated the manuscript with the required information and explanations regarding the experimental design and methodologies used, as requested. Furthermore, they have provided sound and valid counterarguments to the comments expressed in the introduction and discussion sections, and have rightly added supplementary information, allowing for a better elaboration of the data obtained and a clearer interpretation of the results.

Reply: Thank you for your positive evaluation of the revisions. Your feedback reinforces the importance of the review process, and we are pleased that the revised manuscript now offers a clearer interpretation of the results and a more robust presentation of the study.

In light of the comments and revisions made, I support the acceptance of the manuscript. However, I ask the authors to further check and clarify a few aspects that have emerged after a careful assessment of the data provided in the supplementary materials, which were not previously presented. I consider this further minor revision a crucial step towards ensuring the scientific validity and consistency of the study, and ultimately necessary for acceptance.

Reply: Thank you for your supportive feedback. We appreciate your careful assessment of the supplementary materials and agree that addressing these remaining points will improve the manuscript. In the revised version, we have thoroughly checked the supplementary data and clarified all aspects highlighted in your comments.

Materials and Methods

Data analysis

Line 188 and S1 Table: Firstly, thank you for providing this table. The authors mentioned that no data standardisation or normalisation was performed. Clarification of the values presented in the tables is required. Is this combined matrix obtained by merging species abundances with functional categories? Such merging would provide more reliable quantitative information regarding the functional aspects of the community. Otherwise, this table appears to provide only qualitative information related to the presence of each species or taxon and their relative affinity for certain trait modalities, without weighting by the number of individuals (i.e. abundance) found. Consequently, the values are likely to be low in dispersion, and there may be no need for standardisation or normalisation. In my opinion, a quantitative trait analysis would be more informative in the context of this study, and I would suggest considering this type of analysis, as it could add robustness to the work. I would like to clarify that the procedure performed by the authors is correct, but it may be less informative and contextualised, since variations in species abundances are primarily indicative of changes in community structure when exposed to disturbance. If the authors wish to consider my suggestion to obtain a new merged species-trait matrix weighted by abundance, it is necessary to ensure consistency in the calculation of other functional metrics, including CWM. In this case, standardisation or normalisation may also be required. For example, the high abundance of the amphipod Paracorophium excavatum may not be comparable with the lower abundances of other species (e.g. Paradoneis lyra).

Reply: Thank you for raising this point. The values presented in Table S1 are only the fuzzy coding applied to each taxon. Yet, for all our statistical analyses we used the suggested quantitative data obtained from combining the macrobenthic abundances with their associated functional traits. This approach ensures that the analysis incorporates quantitative information on the functional aspects of the community rather than relying solely on presence/absence data. This step was performed using the FD package in R software, which performs the combining step prior to providing the outcomes used in our analysis. This information is presented in lines 199-200. In the revised manuscript, we have added new text to further clarify the process, lines 200-202.

Thank you also for your suggestions regarding the standardization/normalization of the data. While we appreciate the consideration of standardization or normalization, we determined that these steps were not necessary for our dataset. The combined macrobenthic–trait matrix weighted by abundance did not exhibit overdispersion (shade plots), and applying standardization would have artificially reduced the true variability among treatments, potentially masking ecologically meaningful differences. Our approach preserves the natural variation in macrobenthic abundances and trait expression, which is critical for accurately interpreting functional responses to heat stress.

Lines 205-209: It is stated that “Euclidean distance” was used for both structural metrics (i.e. abundances, S, and H’) and functional metrics (FRic, FEve, FDis, FR, and CWM). However, as the functional trait analysis includes both categorical traits (e.g. feeding mode, movement method, etc.) and continuous traits (e.g. body size), the literature recommends using the Gower dissimilarity index (Gower, 1971; Teichert et al., 2017). Euclidean distance is typically applied only to continuous (environmental) variables.

Reply: Thank you for your comment. We acknowledge the recommendation to use Gower dissimilarity when combining categorical and continuous traits. However, in our analysis, we used Euclidean distance because the data input for PERMANOVA consisted of single-variable metrics (e.g., FRic, FEve, FDis, FR, CWM) derived from the combined species–trait matrix, rather than raw trait data mixing categorical and continuous variables. This approach aligns with the protocols suggested by Anderson (2001) for PERMANOVA, where Euclidean distance is commonly applied to univariate or single-metric datasets in ecological studies. Using Euclidean distance ensures consistency with these established practices and is appropriate for the type of data analyzed in this study.

Results

Lines 299-301: The FRic values are surprisingly high given the number of species found (25). A high FRic indicates a very wide trait space, and this index is usually closely linked to species richness.

Reply: Thank you for highlighting this point. We have crosschecked the values and calculations and are correct. The high FRic values observed are due to the way Functional Richness is calculated: FRic considers not only taxa richness but also the distribution of taxa abundances within the multidimensional trait space. Because our analysis uses abundance-weighted trait data, taxa with higher abundances contribute more strongly to the expansion of the trait space, resulting in higher FRic values even with a moderate number of taxa.

We did identify the wrong citation in text of a supplementary table “S5”, we have changed to appropriate supplementary table “S6”, line 302.

Lines 304-305: Please verify that the FEve values exactly match those provided in Table S6. (FEve values range from 0 to 1.)

Reply: Thank you for pointing this out. You are correct that the FEve values in the text did not match those in Table S6. In this revised version of the manuscript, we have carefully reviewed and corrected the values to ensure consistency with the supplementary table, lines 307-308.

REVIEWER #2:

The manuscript has experienced a thorough revision after its initial submission and the authors have addressed all remarks and I particularly appreciate the effort invested into the data provision and strengthening the ecological context of the study. Most of the issues have been sufficiently resolved, but a few comments remain. The referred line numbers below are according to the track change document.

Reply: Thank you for your positive assessment of the revisions and for acknowledging the effort invested in improving the manuscript. We are pleased that the changes have strengthened the ecological context and addressed most of the previous concerns. In this revised version, we have addressed each of the suggestions.

Lines 61: Please clarify these contrasting statements. Line 61: “Estuarine macrobenthic organisms are likely well adapted to cope with extreme weather events such as MHWs …” and line 69: “… these organisms are predicted to be more sensitive to temperature extremes”

Reply: Thank you for highlighting this point. We have revised the text to clarify the apparent contradiction. In the previous revision we added text explaining that “while intertidal macrobenthic communities are generally adapted to natural fluctuations such as tidal cycles and seasonal changes, they remain highly sensitive to extreme and prolonged stressors beyond their usual environmental range, such as marine heatwaves”. In this revised version of the manuscript, we have reviewed the text and provided further clarification: “Estuarine macrobenthic organisms have the potential to cope with extreme weather events such as MHWs for short periods [30-32], however, negative impacts may occur when organisms face prolonged periods of elevated temperature that exceed their tolerance levels [31, 33-35].” and “These organisms are predicted to be more sensitive to temperature as global warming continues to increase, particularly when the temperature extremes are prolonged [36-39].”. Lines 56-57 and 64-65.

Lines 165 (Macrobenthic fauna sampling): Please add the sampling dates for the macrofauna. The current description is confusing due to inconsistencies between the manuscript and the revision responses. Specifically, the manuscript describes experimental treatment days, while the responses alternatively state that (i) Control and Long (7-day exposure) were sampled after 7 days, with the Short (5-day exposure) treatment sampled after 5 days, and (ii) that all treatments were sampled simultaneously after the 7-day experimental period. Please clarify.

Reply: Thank you for highlighting this concern. As suggested by the reviewer, in this revised version of the manuscript we have added the specific sampling dates for the macrofauna to remove any ambiguity, lines 160-161. To clarify, the Control and Long treatments (7-day exposure) were sampled on Day 7, while the Short treatment (5-day exposure) was sampled on Day 5.

Lines 242–249 and Fig. 2: Please clarify whether temperature differences between the Short and Long treatments were statistically tested. In addition, it is difficult to identify from Fig. 2 when the OTC chambers were deployed during the Short treatment, as temperatures appear consistently higher than the Control across all days. If the Short treatment was sampled at the same time as the other treatments (23/02/2024?) – Would you have an explanation why the temperature was elevated on 17 and 18/02/2024?

Reply: Thank you for this comment. Yes, temperature differences between the Control, Short and Long treatments were statistically tested. The details of these analyses are provided in the Methods section (lines 176–182), and the relevant PERMANOVA results are presented in the Results section (lines 236-237). Additionally, we confirm that the data shown in Figure 2 (and in general the data analyzed) correspond only to the periods when the OTC chambers were deployed. Furthermore, as stated in the previous comment, short treatment was not sampled at the same time as the other treatments.

Lines 394–403: While the example of functional traits (e.g. suspension feeders, predators, scavengers) in the Results section (lines 287–291) is appropriate, caution is needed when interpreting and linking these traits, as the discussion establishes an apparent link between them that is not directly supported by the data. To my knowledge, a large body of literature indicates that a common response of benthic invertebrates to excessive heat is deeper burrowing into the sediment (agreeing with your suggestion in line 377), rather than surfacing, unless driven by hypoxia, which I don’t suppose to be the case here. The manuscript suggests surfacing of animals and links this to a trophic cascade of increased resource availability for scavengers feeding on carrion and nutrient cycling. Given the lack of direct observations or data on animal movement in this study, such an interpretation is not sufficiently supported and should be reconsidered or more cautiously framed. In addition, given the availability of data now, I ask the authors to re-examine Table S5. The reported trait proportions (3% in Control, 5% in Short, and 6.8% in Long) appear to correspond to predators rather than scavengers (0.4%, 0.8%, 0.8%, respectively). Please verify this point and revise the text and discussion accordingly.

Reply: Thank you for highlighting this important concern. We agree that, given the lack of direct observations or behavioral data in our study, the interpretation regarding surfacing and its link to trophic cascades should be presented more cautiously. In the revised manuscript, we have reframed this section to clarify that these ideas are speculative and based on patterns reported in previous literature rather than direct evidence from our experiment. Specifically, we now state that while heat stress can influence resource availability and trophic interactions, the potential mechanisms such as surfacing or increased scavenging, remain hypothetical in this context and require further investigation. This adjustment ensures that our discussion acknowledges the limitations of the data while still providing ecological context supported by relevant references. Lines 393-396.

We also thank you for drawing our attention to re-examine Table S5. After carefully re-examining Table S5, we identified an error: the trait modality omnivore was inadvertently omitted, which caused all other feeding trait modality values to be shifted. We have corrected this mistake, and the updated Table S5 now includes the complete set of feeding trait modalities. The revised values align with those described in the text and discussion.

Line 413: According to Mouillot et al. (2013, “A functional approach reveals community responses to disturbances”), FDis is generally expected to decrease under disturbance due to the loss of functions. In contrast, in this study, FDis significantly increases. The authors may wish to contextualize these results – Could the observed pattern be related to the openness of the study system?

Reply: Thank you for this insightful comment. We agree that the observed increase in FDis under disturbance contrasts with the general expectation outlined by Mouillot et al. (2013). In Mouillot et al. (2013) paper is explained that many disturbances select against particular trait values and often reduce functional dispersion (FDis) through loss of functional type. However, FDis can increase in systems that are open to immigration (allowing colonisation by species with novel trait combinations), where disturbance intensity, duration, or spatial scale promote coexistence among complementary trait types, or when opportunistic/invasive species with novel traits enter the community. This pattern has been reported in other open systems where functional redundancy and trait variability buffer against disturbance effects. In this revised version of the manuscript, we have added text to contextualize our unexpected increase in FD

---

## [Editor Report · Decision Letter 2]

28 Dec 2025

Redundancy of macrobenthic functional traits boosts resilience to a simulated heatwave

PONE-D-25-44021R2

Dear Dr. Lam-Gordillo,

We’re pleased to inform you that your manuscript has been judged scientifically suitable for publication and will be formally accepted for publication once it meets all outstanding technical requirements.

Kind regards,

Marcos Rubal García, PhD

Academic Editor

PLOS One
---

## [Editor Report · Acceptance letter]

PONE-D-25-44021R2

PLOS One

Dear Dr. Lam-Gordillo,

I'm pleased to inform you that your manuscript has been deemed suitable for publication in PLOS One. Congratulations! Your manuscript is now being handed over to our production team.

Kind regards,

on behalf of

Dr. Marcos Rubal García

Academic Editor

PLOS One